# A Non-parametric Direct Learning Approach to Heterogeneous Treatment Effect Estimation under Unmeasured Confounding

**Xinhai Zhang**[*]
xzhan222@binghamton.edu

**Xingye Qiao**[*]
xqiao@binghamton.edu

## Abstract

In many social, behavioral, and biomedical sciences, treatment effect estimation is a crucial step in understanding the impact of an intervention, policy, or treatment. In recent years, an increasing emphasis has been placed on heterogeneity in treatment effects, leading to the development of various methods for estimating Conditional Average Treatment Effects (CATE). These approaches hinge on a crucial identifying condition of no unmeasured confounding, an assumption that is not always guaranteed in observational studies or randomized control trials with non-compliance. In this paper, we proposed a general framework for estimating CATE with a possible unmeasured confounder using Instrumental Variables. We also construct estimators that exhibit greater efficiency and robustness against various scenarios of model misspecification. The efficacy of the proposed framework is demonstrated through simulation studies and a real data example.

## 1 Introduction

In various domains, different subjects may exhibit different responses to the same set of treatments. The exploration of this heterogeneity in the effects resulting from exposure has gained substantial interest in recent years. For instance, inferring the heterogeneous effect of a medical treatment on clinical outcome can contribute to the development of personalized treatment (Cai et al., 2011). A similar concept has found application in personalized marketing as well (Chandra et al., 2022). The heterogeneity among subjects can be measured by the disparity in conditional mean outcomes given other covariates, typically referred to as the Conditional Average Treatment Effect (CATE). Another problem closely related to the heterogeneity in treatment effects is the optimal Individualized Treatment Regime (ITR), which is a decision rule that selects treatments for individuals to maximize the expected outcome.

There has been significant development in the literature regarding the estimation of CATE and the optimal ITR in the case of no unmeasured confounding. For example, Q-learning (Qian and Murphy, 2011) models the conditional mean outcome under each treatment separately and the estimated CATE is constructed using the difference between the estimated conditional mean outcomes. The success of this method relies on the correct specification of the outcome models. To address this issue, direct learning (DL) (Tian et al., 2014; Qi and Liu, 2018) and robust direct learning (RD) (Meng and Qiao, 2022) models the conditional contrast between treatments directly, which has been shown to be more robust to model misspecification. Another strand of work approaches with tree-based or forest-based methods. Hill (2011) and Green and Kern (2012) extended the Bayesian Additive Regression Tree (BART) method of Chipman et al. (2010) for estimating heterogeneous treatment effect. Athey and Imbens (2016) proposed Causal Trees with an "honest" splitting approach, wherein the partitioning is

---

[*]Department of Mathematics and Statistics, State University of New York at Binghamton, Binghamton, NY, 13902, USA.

38th Conference on Neural Information Processing Systems (NeurIPS 2024).

constructed in one sample, and the treatment effects within each node are estimated using another sample. This methodology is subsequently adopted in Causal Forest (Wager and Athey, 2018), which extends the random forest algorithm to estimate heterogeneous treatment effects. On the other hand, optimal ITR estimation aims to determine the optimal decision rule for treatment assignment based on subjects' covariates to maximize the mean outcome. A significant line of work in the field involves transforming ITR estimation into a classification problem through the use of Inverse Probability Weighting (IPW). Notable contributions include Outcome Weighted Learning (OWL) (Zhang et al., 2012; Zhao et al., 2012) and Residual Weighted Learning (RWL) (Zhou et al., 2017).

The aforementioned methods all rely on the key assumption of no unmeasured confounding to identify the heterogeneous treatment effect or the optimal ITR. However, this assumption is in most cases unverifiable (if not untrue) in observational studies or randomized controlled trials (RCT) with non-compliance. A well-known approach that takes into account the unmeasured confounding is the use of an instrumental variable (IV). A proper IV is usually a pre-treatment variable that is independent of any possible unmeasured confounder while correlated with the treatment. For example, in RCT with non-compliance, the random treatment assignment can be considered as an IV while the treatment received is considered the treatment variable. Here these two are clearly correlated since a subject will not receive the treatment if they are not assigned one, though the strength of the correlation may depend on other characteristics such as the education level of the subject.

There is a growing literature on estimating heterogeneous treatment effects or optimal ITR under unmeasured confounding using IV. Imbens and Angrist (1994) identified and estimated the so-called Local Average Treatment Effect (LATE), restricted to the subgroup of the always-compliant population, with the help of an IV. More recently, machine learning methods like Doubly Robust IV (Syrgkanis et al., 2019) and Generalized Random Forest (Athey et al., 2019) have shown their applicability and effectiveness in various settings including unmeasured confounding, particularly when used in conjunction with an IV. Wang and Tchetgen Tchetgen (2018) introduced two alternative assumptions on the unobserved confounders and the IV, which enable the identification of the Average Treatment Effect (ATE). They proposed an estimator that has the so-called multiply robustness property, which guarantees consistent estimate under three observed data models. These findings were incorporated into Cui and Tchetgen Tchetgen (2021) to obtain an optimal ITR estimation while accounting for unmeasured confounding. On the other hand, Frauen and Feuerriegel (2022) utilized these findings for CATE estimation.

In this paper, we propose a new framework for estimating CATE using IV when there exist unmeasured confounders. This framework can be viewed as an extension of the Direct Learning method under unconfoundedness to the case that allows the existence of unmeasured confounding. We call the proposed method Direct Learning using Instrumental Variables (IV-DL). The proposed framework is easy to implement under many flexible learning methods. Additionally, we introduce several efficient and robust estimators by residualizing the outcome. These estimators have been demonstrated to be robust to multiple model misspecification scenarios.

The rest of this paper is organized as follows. The notations and some related preliminaries are introduced in Section 2. The proposed framework IV-DL is formally introduced in Section 3. In Section 4 and 5, we proposed efficient and robust estimators. In Section 6, we conduct simulation studies and compare the performance with existing methods in the literature. A real data example is included in Section 7. Section 8 concludes the paper with a discussion on possible future work. Proofs and additional simulations are provided in the Appendix.

## 2   Notations and Preliminaries

Denote $A \in \mathcal{A} = \{+1, -1\}$ as the binary treatment, and $X \in \mathcal{X} \subseteq \mathbb{R}^p$ the pre-treatment covariates. We adapt the potential outcome framework (Rubin, 1974) in causal inference and denote by $Y(a) \in \mathbb{R}$ the potential outcome that the subject would have obtained if the received treatment was $a \in \mathcal{A}$. The observed outcome is then given by $Y = Y(A) = Y(1)\mathbb{1}[A = 1] + Y(-1)\mathbb{1}[A = -1]$. Denote by $U$ the unobserved confounder of the effect of $A$ on $Y$. Suppose we have access to a pre-treatment binary IV denoted by $Z \in \mathcal{Z} = \{+1, -1\}$. Then the complete data consists of independent and identically distributed copies of $(Y, X, A, U, Z)$, even though only copies of $(Y, X, A, Z)$ are observed.

Our goal is to estimate the Conditional Average Treatment Effect (CATE), defined as $\Delta(x) \triangleq \mathbb{E}[Y(1) - Y(-1)|X = x]$. As mentioned in Section 1, most of the prior works are based on the core assumption of no unmeasured confounding:

**Assumption 1** (Unconfoundedness). $Y(a) \perp\!\!\!\perp A|X$ for $a = \pm 1$.

This assumption essentially implies that the observed covariates $X$ would suffice to account for the confounding of the effect of $A$ on $Y$, thereby excluding the presence of $U$. Under the above assumption of unconfoundedness, it can be easily verified that CATE is identified by $\Delta(x) = \mathbb{E}[Y|A = 1, X = x] - \mathbb{E}[Y|A = -1, X = x]$. Q-learning (Qian and Murphy, 2011) models the two conditional mean outcomes separately and estimates the CATE by taking the difference between these estimates. Consequently, its effectiveness depends on correctly specifying the models for the conditional mean outcomes. Denote the propensity score for the treatment as $\pi_A(a, x) = \mathrm{P}[A = a|X = x]$ for $a = \pm 1$. Direct Learning (Qi and Liu, 2018; Tian et al., 2014) propose to directly model for the heterogeneous treatment effect, based on the observation that $\Delta(x) = \mathbb{E}[AY/\pi_A(A, X)|X = x]$. In other words, one can obtain an estimate of CATE by regressing the modified outcome $AY/\pi_A(A, X)$ on $X$. Robust Direct Learning (RD) Meng and Qiao (2022) further extends this framework by residualizing the outcome using an estimate of the main effect, which is the average of the two conditional mean outcomes. This method demonstrates double robustness in the sense that it yields consistent estimation of CATE if either the propensity score or the main effect is correctly specified. Despite the success in RCT or observational studies, all the methods mentioned above rely on the unconfoundedness Assumption 1. In the next section, we will introduce a general framework that directly models CATE using an IV approach when there exists unmeasured confounding.

## 3  Direct Learning with Instrumental Variable Approach

In this paper, we look beyond Assumption 1, and consider the existence of an unmeasured confounder $U$. To establish the identification of CATE in this setting, we approach with the use of a proper IV. We will start with the following assumptions seen in Cui and Tchetgen Tchetgen (2021).

**Assumption 2.** *This assumption consists of five parts as follows:*

    *a.* $Y(z, a) \perp\!\!\!\perp (Z, A)|X, U$ *for* $z, a = \pm 1$.

    *b.* $Z \not\!\perp\!\!\!\perp A|X$.

    *c.* $Z \perp\!\!\!\perp U|X$.

    *d.* $Y(z, a) = Y(z', a)$ *for* $z, z', a = \pm 1$.

    *e.* $0 < \pi_Z(1, X) < 1$ *almost surely, where* $\pi_Z(z, x) = \mathrm{P}[Z = z|X = x]$ *for* $z = \pm 1$.

Here, $Y(z, a)$ represents the potential outcome that would be observed if a subject were exposed to treatment $a \in \mathcal{A}$, and the IV takes a value of $z \in \mathcal{Z}$. Assumption 2.a rules out the existence of any other confounder, except for $X$ and $U$, for the joint effect of $Z$ and $A$ on the outcome $Y$. However, this unconfoundedness is hidden from the data collected, since $U$ is never observed. Assumptions 2.b-2.e provides us with a well-defined IV. Assumption 2.b requires a correlation between the IV and the treatment given observed covariates. In many applications, a strong correlation is often necessary to ensure accurate inference in the estimation process. Assumption 2.c guarantees that the causal effect of $Z$ on $Y$ is not confounded given $X$; otherwise $Z$ suffers the same issue as $A$. Additionally, required by Assumption 2.d, the causal effect of $Z$ on $Y$ can only be mediated by the treatment $A$. In light of this assumption, we omit the argument $z$ in the potential outcome and denote the common value as $Y(a)$. Assumption 2.e implies that each subject has a positive chance of having either value of the IV. An example of the relationships between variables that satisfy Assumption 2 is presented in a directed acyclic graph in Figure 1. In order to identify the CATE, we also need the following assumption on the unmeasured confounder.

**Assumption 3.** *At least one of the following is true:*

    *a.* $\mathbb{E}[A|Z = 1, X, U] - \mathbb{E}[A|Z = -1, X, U] = \mathbb{E}[A|Z = 1, X] - \mathbb{E}[A|Z = -1, X]$

    *b.* $\mathbb{E}[Y(1) - Y(-1)|X, U] = \mathbb{E}[Y(1) - Y(-1)|X]$

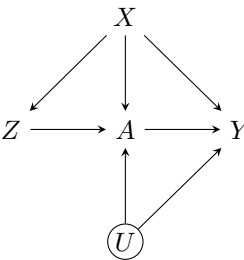

Figure 1: A directed acyclic graph with unmeasured confounding and an IV

Assumption 3 states that, conditional on the measured covariates, either the additive effect of $Z$ on $A$ is independent of $U$, or the additive effect of $A$ on $Y$ is independent of $U$. Now, we finally have identification of the CATE.

**Proposition 1.** *Under Assumptions 2–3, the CATE can be identified by*

$$\Delta(x) = \frac{\mathbb{E}[Y|Z=1, X=x] - \mathbb{E}[Y|Z=-1, X=x]}{\mathrm{P}[A=1|Z=1, X=x] - \mathrm{P}[A=1|Z=-1, X=x]} \tag{1}$$

$$= \mathbb{E}\left[\frac{ZY}{\delta(x)\pi_Z(Z,x)}\bigg| X=x\right], \tag{2}$$

*where* $\pi_Z(z,x) = \mathrm{P}[Z=z|X=x]$ *and* $\delta(x) = \mathrm{P}[A=1|Z=1, X=x] - \mathrm{P}[A=1|Z=-1, X=x]$ *for any* $x \in \mathcal{X}$.

The first equality (1) was shown in Wang and Tchetgen Tchetgen (2018), which means that the CATE is identified by the conditional Wald estimand. Equation (2) reveals an interesting observation that we do not need the realized treatment $A$ as long as we have $\delta(x)$, which can be viewed as the conditional effect of the IV on the treatment given observed covariates. Hereafter, we denote the conditional means of $Y$ and $A$ by $\mu_z^Y(x) = \mathbb{E}[Y|Z=z, X=x]$ and $\mu_z^A(x) = \mathbb{E}[A|Z=z, X=x]$, respectively, for any $z \in \{-1, +1\}$ and $x \in \mathcal{X}$.

### 3.1 Conditional Average Treatment Effect Estimation

In this section, we will introduce the IV-DL framework. Motivated by Equation (2), the next lemma offers a way to estimate $\Delta(x)$ using inverse propensity score of IV as weight.

**Lemma 1.** *Under Assumptions 2–3,*

$$\Delta \in \operatorname*{argmin}_{f} \mathbb{E}\left[\frac{1}{\pi_Z(Z,X)}\left(\frac{2ZY}{\delta(X)} - f(X)\right)^2\right].$$

Based on Lemma 1, we can adopt many existing regression methods to obtain an estimate on CATE by regressing the modified outcome on the covariates, weighted by the propensity score for $Z$. Specifically, given the data $\{y_i, x_i, a_i, z_i\}_{i=1}^n$, an estimator $\hat{\pi}_Z$ of the propensity score function and an estimator $\hat{\delta}$ of the effect of $Z$ on $A$,, the IV-DL estimate for $\Delta$ is given by

$$\hat{f}(x) = \operatorname*{argmin}_{f \in \mathcal{F}} \frac{1}{n}\sum_{i=1}^{n} \frac{1}{\hat{\pi}_Z(z_i, x_i)}\left(\frac{2z_i y_i}{\hat{\delta}(x_i)} - f(x_i)\right)^2 + \lambda\|f\|_{\mathcal{F}},$$

where $\mathcal{F}$ is a function space with norm $\|\cdot\|_{\mathcal{F}}$, and $\lambda \geq 0$ is the tuning parameter for the regularization term $\|f\|_{\mathcal{F}}$. To obtain $\hat{\pi}_Z$, we can fit a logistic regression of $Z$ on $X$ or a non-parametric model such as random forest. Since $\delta$ is the treatment effect of $Z$ on $A$, it is noteworthy that, under Assumption 3, estimation of $\delta$ can be viewed as a CATE estimation problem with unconfoundedness. In this case, $A$ may be viewed as a binary "outcome" and $Z$ a binary "treatment". Thus, we can adopt many existing CATE estimation methods such as Q-learning, DL, and Causal Forest.

The proposed framework allows a variety of learning methods to model the treatment effect $\Delta(x)$. For example, under the linear model, we may model $f(x) = \tilde{x}^T\boldsymbol{\beta}$ where the regression coefficients

are $\boldsymbol{\beta}$ and $\tilde{x}_i \triangleq (1, x_i^T)^T$. Then IV-DL estimator for $\boldsymbol{\beta}$ is

$$\hat{\boldsymbol{\beta}} = \operatorname*{argmin}_{\boldsymbol{\beta} \in \mathbb{R}^{p+1}} \frac{1}{n} \sum_{i=1}^{n} \frac{1}{\hat{\pi}_Z(z_i, x_i)} \left( \frac{2z_i y_i}{\hat{\delta}(x_i)} - \tilde{x}_i^T \boldsymbol{\beta} \right)^2$$

and the CATE $\Delta(x)$ is estimated by $\hat{f}(x) = \tilde{x}^T \hat{\boldsymbol{\beta}}$.

In high dimensional setting where $p$ is large, sparse regularization can be easily applied here because the optimization is essentially a weighted least square problem. For example, we can use Least Absolute Shrinkage and Selection Operator (LASSO) and the estimator of $\boldsymbol{\beta}$ is given by

$$\hat{\boldsymbol{\beta}}^{lasso} = \operatorname*{argmin}_{\boldsymbol{\beta} \in \mathbb{R}^{p+1}} \frac{1}{n} \sum_{i=1}^{n} \frac{1}{\hat{\pi}_Z(z_i, x_i)} \left( \frac{2z_i y_i}{\hat{\delta}(x_i)} - \tilde{x}_i^T \boldsymbol{\beta} \right)^2 + \lambda \|\boldsymbol{\beta}\|_1,$$

where $\lambda > 0$ is the tuning parameter for the $l_1$ penalty.

In practice, there is no guarantee that the true treatment effect follows a linear model. For a more complex model, we can adopt nonlinear methods such as Kernel Ridge Regression (KRR) and solve

$$\operatorname*{argmin}_{\boldsymbol{\beta} \in \mathbb{R}^n, \beta_0 \in \mathbb{R}} \frac{1}{n} \sum_{i=1}^{n} \frac{1}{\hat{\pi}_Z(z_i, x_i)} \left( \frac{2z_i y_i}{\hat{\delta}(x_i)} - (\boldsymbol{K}_i^T \boldsymbol{\beta} + \beta_0) \right)^2 + \lambda \boldsymbol{\beta}^T \mathbf{K} \boldsymbol{\beta},$$

where $\boldsymbol{K}_i$ is the $i$-th column of the kernel matrix $\mathbf{K} = (K(x_i, x_j))_{n \times n}$ and $K(\cdot, \cdot)$ is a kernel function. KRR might be computationally expensive when dealing with large datasets. In such cases, other machine learning methods capable of solving a weighted least squares problem can be considered. Examples include local regression, regression trees, random forests, and neural networks.

## 3.2 Optimal Individualized Treatment Regime Estimation

In some domains, the optimal Individualized Treatment Regime (ITR) can be of interest. The goal here is to find a mapping $d : \mathcal{X} \to \mathcal{A}$ from a specific class $\mathcal{D}$ to maximizes the expected outcome: $d^* \triangleq \operatorname{argmax}_{d \in \mathcal{D}} \mathbb{E}[Y(d(X))]$, where $Y(d(X))$ is the potential outcome that the subject $X$ obtained after receiving treatment $d(X)$, and $\mathbb{E}[Y(d(X))]$ is also known as the Value of the regime $d$.

ITR and CATE are closely related. For example, in the binary treatment setting, the CATE $\Delta$ is the difference between two conditional mean outcomes. Assuming greater values of outcome is preferred, then the sign of $\Delta$ will determine which treatment is optimal. It can be verified that $d^*(x) = \operatorname{sign}(\Delta(x))$. Therefore, we define the estimated optimal ITR using IV-DL as $\hat{d}(x) = \operatorname{sign}(\hat{f}(x))$, where $\hat{f}(x)$ may be any CATE estimator introduced in the last subsection.

## 4 Efficient Estimators by Residualization

In the literature, considerable advancements have been made to enhance the efficiency and robustness of the CATE and optimal ITR estimation. To this end, residualization and augmentation are two common strategies. For example, in the IPW framework for optimal ITR estimation, Zhou et al. (2017) and Zhou and Kosorok (2017) proposed to replace the outcome by its residual $Y - \hat{g}(x)$ in estimation of the optimal regime, where $\hat{g}(x)$ is an estimate of the weighted average of the conditional mean outcomes. For the estimation of CATE, Meng and Qiao (2022) residualized the outcome by an estimate of the average of conditional mean outcomes. Frauen and Feuerriegel (2022) proposed augmenting a preliminary estimate of CATE to enhance the robustness of the estimator.

In this section, we present the Robust Direct Learning using IV approach (IV-RDL), which involves residualizing the outcome in IV-DL to enhance both efficiency and robustness. We propose two ways of residualization, referred to as IV-RDL1 and IV-RDL2, respectively. They are shown to reduce the variance when estimating CATE. In Section 5, we show that they have robustness properties when confronted with model misspecification for nuisance variables.

## 4.1 Residualization using a Function of Covariates

We first consider residualizing the outcome by a function of the observed covariates only. Ideally, we would like to find a function $g : \mathcal{X} \to \mathbb{R}$ that can improve the efficiency of the estimation on CATE,

while keeping it consistent. As shown in the following lemma, the consistency of the estimator is in fact preserved under a shift of $Y$ by any function of the observed covariates.

**Lemma 2.** *For any measurable $g : \mathcal{X} \to \mathbb{R}$ and any probability distribution for $(Y, X, A, Z)$*

$$\Delta \in \operatorname*{argmin}_{f} \mathbb{E}\left[\frac{1}{\pi_Z(Z, X)}\left(\frac{2(Y - g(X))Z}{\delta(X)} - f(X)\right)^2\right]$$

Asymptotically, the variance of the estimator is related to the variance of the derivative of $[\pi_Z(Z, X)]^{-1}[2(Y - g(X))Z Z \delta^{-1}(X) - f(X)]^2$, the weighted loss for each individual. Hence, it is natural to choose $g$ that minimize the variance of $[\pi_Z(Z, X)]^{-1}[2(Y - g(X))Z Z \delta^{-1}(X) - f(X)]$. See Appendix B for a more detailed discussion using the linear model as an example. The following theorem gives us the minimizer.

**Theorem 1.** *Among all measurable $g : \mathcal{X} \to \mathbb{R}$, the following function minimize the variance of $\frac{1}{\pi_Z(Z,X)}\left(\frac{2(Y - g(X))Z}{\delta(X)} - f(X)\right)$:*

$$g^*(x) \triangleq \frac{1}{2}\mathbb{E}\left[\frac{Y}{\pi_Z(Z, X)}\bigg| X = x\right] = \frac{\mu_1^Y(x) + \mu_{-1}^Y(x)}{2}. \tag{3}$$

There is an interesting interpretation of the optimal function $g^*$, which equals the average of $\mu_1^Y(x)$ and $\mu_{-1}^Y(x)$. Recall that Eq. (1) states that CATE under unmeasured confounding is identified by the ratio of two contrasts, where the numerator happens to be $\mu_1^Y(x) - \mu_{-1}^Y(x)$. The residualization strategy amounts to shifting the outcome $Y$, and hence $\mu_1^Y(x)$ and $\mu_{-1}^Y(x)$ as well. Naturally, shifting both by their average will not affect their difference, but it will reduce the variance. A similar residualization was incorporated in RD under unconfoundedness (Meng and Qiao, 2022), where the goal was to learn the contrast between conditional mean outcomes given the two treatments.

In practice, $g^*$ needs to be estimated before we can estimate the CATE. There are several approaches to obtain the estimate of $g^*$, denoted by $\hat{g}^*$. For example, we can take the average of estimated conditional mean outcomes, i.e., $\hat{g}^*(x) = (\hat{\mu}_1^Y(x) + \hat{\mu}_{-1}^Y(x))/2$. One can also regress $Y/(2\pi_Z(Z, X))$ on $X$, inspired by Eq. (3). Given $\hat{g}^*(x)$, the IV-RDL1 estimator for $\Delta$ is obtained by

$$\hat{f}_g(x_i) = \operatorname*{argmin}_{f \in \mathcal{F}} \frac{1}{n}\sum_{i=1}^{n}\frac{1}{\hat{\pi}_Z(z_i, x_i)}\left(\frac{2(y_i - \hat{g}^*(x_i))z_i}{\hat{\delta}(x_i)} - f(x_i)\right)^2 + \lambda\|f\|_{\mathcal{F}}.$$

In Section 5, we will show that this estimator is robust against misspecification of either $g^*$ or $\pi_Z$, given that $\delta$ is correctly specified.

### 4.2 Residualization using Covariates, Treatment, and IV

In this paper, we also consider an alternative way of residualizing the outcome by a function $h : (\mathcal{X}, \mathcal{A}, \mathcal{Z}) \to \mathbb{R}$. Like IV-RDL1, the optimal choice is the function that minimizes the variance while maintaining the consistency of CATE estimation. Among all functions that still convey consistent CATE estimation, the following three equivalent functions minimize the variance of $[\pi_Z(Z, X)]^{-1}[2(Y - h(X, A, Z))Z Z \delta^{-1}(X) - f(X)]$.

$$h_1^*(x, a, z) = \mu_1^Y(x) + \Delta(x)\big(a - \mu_1^A(x) - z\delta(x)\big)/2$$
$$h_2^*(x, a, z) = \mu_{-1}^Y(x) + \Delta(x)\big(a - \mu_{-1}^A(x) - z\delta(x)\big)/2$$
$$h_3^*(x, a, z) = m^Y(x) + \Delta(x)\big(a - m^A(x) - z\delta(x)\big)/2$$

where $m^Y(x) \triangleq (\mu_1^Y(x) + \mu_{-1}^Y(x))/2$ and $m^A(x) \triangleq (\mu_1^A(x) + \mu_{-1}^A(x))/2$. The technical details are provided in the Appendix C. In practice, all these conditional means ($\mu_{-1}^Y$, $\mu_1^Y$, $\mu_{-1}^A$ and $\mu_1^A$) need to be estimated, together with estimations of $\pi_Z$ and $\delta$. Additionally, we need to obtain a preliminary estimate of CATE. The IV-RDL2 estimator is constructed by,

$$\hat{f}_h(x_i) = \operatorname*{argmin}_{f \in \mathcal{F}} \frac{1}{n}\sum_{i=1}^{n}\frac{1}{\hat{\pi}_Z(z_i, x_i)}\left(\frac{2(y_i - \hat{h}^*(x_i, a_i, z_i))z_i}{\hat{\delta}(x_i)} - f(x_i)\right)^2 + \lambda\|f\|_{\mathcal{F}},$$

where $\hat{h}^*$ is an estimator for one of $h_1^*$, $h_2^*$ and $h_3^*$.

# 5 Robustness Properties

In this section, we investigate the robustness properties of IV-RDL1 and IV-RDL2. We start with the following theorem to demonstrate the double robustness property of the IV-RDL1 that residualizes the outcome by using $g(x)$.

**Theorem 2.** *Suppose Assumption 2–3 holds, and we have a consistent estimator of $\delta$, denoted by $\hat{\delta}$. Let $\tilde{\pi}_Z$ be a working model for $\pi_Z$, and $\tilde{g}$ be a working model for $g^*$. Then we have*

$$\Delta \in \underset{f \in \{\mathcal{X} \to \mathbb{R}\}}{\arg\min} \; \mathbb{E}\left[\frac{1}{\tilde{\pi}_Z(Z, X)}\left(\frac{(Y - \tilde{g}(X))Z}{\hat{\delta}(X)} - f(X)\right)^2\right]$$

*if either $\tilde{\pi}_Z(z, x) = \pi_Z(z, x)$ or $\tilde{g}(x) = g_1^*(x)$ almost surely.*

Theorem 2 indicates that we will have a doubly robust estimator for $\Delta$ if either $\pi_Z$ or $g^*$ is correctly specified when $\delta$ is known or correctly specified. However, the requirement of a consistent estimate of $\delta$ would not pose a significant issue in practical application, since it is essentially a CATE estimation problem under no unmeasured confounding. A consistent estimator for $\delta$ can be found by implementing any state-of-the-art CATE estimation method in the literature.

For the IV-RDL2, there are more nuisance variables that need to be estimated. The next theorem shows that IV-RDL2 is robust to various scenarios of misspecified nuisance variables.

**Theorem 3.** *Suppose Assumption 2–3 holds. Let $\tilde{\pi}_Z$, $\tilde{\delta}$, $\tilde{\mu}_1^Y$, $\tilde{\mu}_{-1}^Y$, $\tilde{\mu}_Z^A$, $\tilde{\mu}_{-1}^A$, $\tilde{m}^Y$, $\tilde{m}^A$ and $\tilde{\Delta}$ be working models for $\pi_Z$, $\delta$, $\mu_1^Y$, $\mu_{-1}^Y$, $\mu_Z^A$, $\mu_{-1}^A$, $m^Y$, $m^A$ and $\Delta$, respectively. Denote $\tilde{h}_1$, $\tilde{h}_2$ and $\tilde{h}_3$ as chosen augmentation formulated according to $h_1^*$, $h_2^*$ and $h_3^*$ using working estimates. Then we have*

$$\Delta \in \underset{f \in \{\mathcal{X} \to \mathbb{R}\}}{\arg\min} \; \mathbb{E}\left[\frac{1}{\tilde{\pi}_Z(Z, X)}\left(\frac{(Y - \tilde{h}(X, A, Z))Z}{\hat{\delta}(X)} - f(X)\right)^2\right]$$

*if any one of the following condition is satisfied: (1) $\tilde{\pi}_Z = \pi_Z$ and $\tilde{\Delta} = \Delta$ almost surely, and $\tilde{h}$ can be any one of $\tilde{h}_1$, $\tilde{h}_2$ and $\tilde{h}_3$. (2) $\tilde{\pi}_Z = \pi_Z$ and $\tilde{\delta} = \delta$ almost surely, and $\tilde{h}$ can be any one of $\tilde{h}_1$, $\tilde{h}_2$ and $\tilde{h}_3$. (3) $\tilde{\mu}_1^Y = \mu_1^Y$, $\tilde{\mu}_1^a = \mu_1^A$ and $\tilde{\Delta} = \Delta$ almost surely, and $\tilde{h} = \tilde{h}_1$. (4) $\tilde{\mu}_{-1}^Y = \mu_{-1}^Y$, $\tilde{\mu}_{-1}^a = \mu_{-1}^A$ and $\tilde{\Delta} = \Delta$ almost surely, and $\tilde{h} = \tilde{h}_2$. (5) $\tilde{m}^Y = m^Y$, $m^A = m^A$ and $\tilde{\Delta} = \Delta$ almost surely, and $\tilde{h} = \tilde{h}_3$. (6) $\tilde{m}^Y = m^Y$, $m^A = m^A$ and $\tilde{\delta} = \delta$ almost surely, and $\tilde{h} = \tilde{h}_3$.*

Theorem 3 summarizes in total six cases of the minimal combination of correctly specified nuisance variables in order to have a consistent estimate of CATE. The three choices of residualization functions possess robustness against different scenarios. In the first two scenarios, obtaining a consistent estimate of CATE is guaranteed as long as we correctly specify $\pi_Z$ and either $\Delta$ or $\delta$. This consistency holds irrespective of the choice of the three $\tilde{h}$ functions. In practice, the second scenario may be particularly accessible, especially when $\pi_Z$ is known. The other scenarios are less likely to be verified in practice and therefore requires more domain knowledge of the data structure. Specifically, scenarios (3)-(5) requires the corresponding set of conditional means to be correctly specified as well as the preliminary $\Delta$. Lastly, in scenario (6), when $\delta$ and the averages of conditional means are correctly specified, the IV-RDL2 will also provide a consistent estimate of CATE.

While working on this paper, we encountered unpublished work by Frauen and Feuerriegel (2022) that is similar to our IV-RDL2 estimator. Inspired by Wang and Tchetgen Tchetgen (2018), Frauen and Feuerriegel introduced the MRIV framework, which is a two-step process. First, a preliminary estimator of CATE and nuisance estimators of $\delta$, $\pi_Z$, $\mu_{-1}^Y$ and $\mu_{-1}^A$ are obtained. Then, a pseudo-outcome is created by augmenting the preliminary CATE with the nuisance estimates, and the final CATE estimator is obtained by regressing the pseudo-outcome on the covariates. As shown in Wang and Tchetgen Tchetgen (2018), this estimator is robust against model misspecification of the nuisance variables in three of the six scenarios in Theorem 3 (scenarios (1), (2), and (4)). Our numerical studies have shown that our proposed IV-DL framework performs better than the MRIV method.

# 6 Simulation Study

In this section, we present the results of the simulation study conducted to assess the performance of the proposed IV-DL framework. We compared the proposed method with Bayesian additive regression trees (BART; Chipman et al., 2010), robust direct learning (RD; Meng and Qiao, 2022), causal forest with IV approach (CF; Athey et al., 2019), MRIV method (Frauen and Feuerriegel, 2022), and weighted learning with IV approach (IPW-MR; Cui and Tchetgen Tchetgen, 2021).

## 6.1 Simulation Settings

We begin by introducing the data-generating mechanism. The covariates, denoted as $X = (X_1, X_2, X_3, X_4, X_5)$, were generated from uniform distribution with $X_i \sim Unif(-1, 1)$ for $i = 1, \ldots, 5$. We followed Cui and Tchetgen Tchetgen (2021) and generated the treatment $A$ under logistic model with probability for success: $\mathbb{P}(A = 1|X, Z, U) = \text{expit}\{2X_1 + 2.5Z - 0.5U\}$, where the instrumental variable $Z$ was a Bernoulli random variable with probability $1/2$ and $U$ was the unobserved confounder that followed Bridge distribution with parameter $\phi = 1/2$. By the results from Wang and Louis (2003), the above usage of Bridge distribution will guarantee that the marginal distribution $f(A|X, Z)$ can be modeled directly by logistic regression. In other words, there exists some vector $\boldsymbol{\alpha}$ such that $logit\{\mathbb{P}(A = 1|X, Z)\} = \boldsymbol{\alpha}^T(1, X, Z)$.

The outcome $Y$ was generated in two different settings corresponding to linear and non-linear models of the true CATE:

1. $Y = h(X) + q(X)A + 0.5U + \epsilon$
2. $Y = h(X) + \{\exp(q(X)) - 1\}A + U + \epsilon$

where the error term $\epsilon$ follows $N(0, 1)$. Functions $h(X)$ and $q(X)$ are defined as follows:

$$h(X) = 0.5 + 0.5X_1 + 0.8X_2 + 0.3X_3 - 0.5X_4 + 0.7X_5$$
$$q(X) = 0.2 - 0.6X_1 - 0.8X_2.$$

In Setting 1, the true CATE is $2q(x)$, which is linear in $x$. In Setting 2, the true CATE is $2(exp(q(x)) - 1)$, which is nonlinear. The sample size for each setting was 500 and the simulation was repeated 100 times. An independent sample of size 5000 was used to evaluate the performance of different methods. The proposed methods were implemented according to Sections 3 and 4 with $\hat{\delta}(X)$ estimated by causal forest ("grf" package) and the other nuisance variables estimated by random forest. For methods that require to estimate the same nuisance variable, they shared the same copies of nuisance estimates.

## 6.2 Numerical Results

We compared all methods based on three performance metrics in the testing sample: the correct classification rate by the estimated ITR (AR); the value function evaluated at the estimated ITR (Value); the mean squared error of the estimated CATE (MSE). Table 1 reports the mean and standard error of these three evaluation metrics over 100 replications for different methods in the two settings.

Table 1: Simulation results: mean$\times 10^{-2}$(SE$\times 10^{-2}$). IPW-MR: the multiply robust weighted learning; BART: Bayesian additive regression trees; RD: robust direct learning; CF: causal forest. The empirical maximum value is 0.998 for setting 1 and 1.01 for setting 2.

|   |       | BART       | RD        | IPW-MR    | CF        | MRIV      | IV-DL     | IV-RDL1       | IV-RDL2    |
|---|-------|------------|-----------|-----------|-----------|-----------|-----------|---------------|------------|
|   | MSE   | 121(3.4)   | 97.6(2.9) | NA        | 89.6(1.8) | 66.3(2.3) | 55.5(3.8) | **40.5(2.9)** | 42.5(3.3)  |
| 1 | AR    | 66.3(0.7)  | 71.4(0.5) | 84.1(0.7) | 79.1(0.7) | 78.3(0.6) | 81.4(1)   | **84.6(0.8)** | 83.7(1)    |
|   | Value | 75.4(0.8)  | 81.6(0.5) | 84.1(0.7) | 85.4(0.7) | 84.5(0.7) | 87.1(1)   | **89.9(0.9)** | 88.9(1.1)  |
|   | MSE   | 449(11.1)  | 397(9.8)  | NA        | 149(2.9)  | 150(5.6)  | 164(8.9)  | **140(7.7)**  | 142(7.4)   |
| 2 | AR    | 57.6(0.6)  | 60.8(0.7) | 55.5(0.3) | 70.1(1)   | 68.8(0.9) | 77.1(0.9) | **77.9(0.8)** | 77.2(0.9)  |
|   | Value | 53.1(1.6)  | 61(1.6)   | 81.6(0.2) | 83.5(1.2) | 81.6(1)   | 89.9(1)   | **90.6(0.9)** | 90(1)      |

Among the methods implemented, BART and RD rely on the unconfoundedness assumption and therefore fail to identify CATE when there is unobserved confounding. Both IPW-MR and CF make use of the IV to take unmeasured confounding into account. IPW-MR had fine performances on estimating ITR and maximizing the value. However, it was not designed to estimate CATE. CF performs slightly worse than IPW-MR in terms of AR and value in Setting 1, despite offering a CATE estimation. Its performance is more competitive compared to IPW-MR in Setting 2. Our proposed methods showed superior performances on all the metrics. In particular, IV-RDL1, which residualized the outcome using averages of the estimated conditional means, outperformed all the methods in both settings. IV-RDL2 had a more complicated residualization, and achieved the second-best performance (but still fairly close to IV-RDL1). Even the unresidualized IV-DL performed better than other methods in most of the metrics. Additional simulation results on testing the robustness of the proposed framework is reported in Appendix D.

## 7 Data Analysis

In this section, following Angrist and Evans (1998), we study the causal effect of child-rearing on a mother's labor-force participation, using a sample of married mothers with two or more children from the U.S. 1980 census data (80PUMS). Assuming the sex of children is random, "first two children mixed sex or not" becomes a suitable instrumental variable for the causal effect of having a third child on a mother's labor force participation. Angrist and Evans showed that having a third child reduces women's labor force participation on average. Our goal is to investigate heterogeneity among families, offering personalized insights on the decision to have a third child and its impact on employment opportunities. We used a dataset of 478,005 subjects with at least two children. The outcome, $Y$, represents whether the mother was employed in the year preceding the census. The treatment, $A$, indicates whether the mother had three or more children at the census time, and the instrumental variable, $Z$, indicates whether the first two children were of the same sex. We considered five covariates, $X$: mother's age at first birth, age at census time, years of education, race, and the father's income.

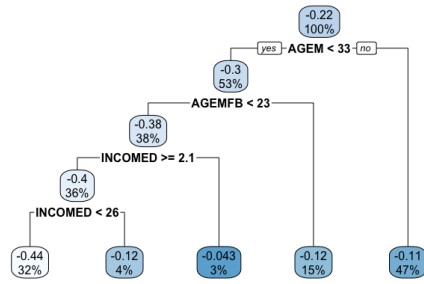

Figure 2: Tree splitting of estimated CATE on covariates. The five leaf nodes shall be numbered 1–5.

Figure 3: Histograms of estimated CATE in three majority subgroups

We used the random forest algorithm for both the implementation of the proposed method and the estimation of the nuisance variables $\mu_z^Y$, $\mu_z^A$, $\delta$, and $\pi_Z$. The preliminary CATE estimator was formulated according to Eq. 1 with plug-in estimates on the conditional means. To identify subgroups with distinct treatment effects, we used the estimated CATE as the response to construct a regression tree, shown in Figure 2. The splits occurred at the mother's age at census (33), age at first birth (23), and father's income ($2.1k/year and $26k/year). By investigating the five subgroups (32%, 4%, 3%, 15%, and 47% of the sample), labeled as groups 1–5, we have made the following observations. First, older mothers are more likely to work after having a third child (subgroups 4 and 5 show a larger estimated treatment effect). Second, younger mothers with very low-income fathers (subgroup 3) tend to stay in the labor force after the third child. Lastly, younger mothers are more likely to stop working if their husband's income is between $2.1k and $26k/year (subgroups 1-3). Figure 3 displays the histogram of estimated CATE for the three majority groups (1, 4, and 5). The estimated CATE for group 1 is overall smaller than for groups 4 and 5. We also constructed 3-dimensional scatter

plots based on the three splitting variables for a more detailed look at the heterogeneity (shown in Appendix E).

## 8  Conclusions

In this paper, we proposed a new framework to estimate CATE under unmeasured confounding by using an instrumental variable. Under the proposed framework, the estimation procedure boils down to solving a weighted least square problem, which can be tackled with any modern statistical or machine learning method. We also constructed two robust estimators by residualizing the outcome, which are shown to be more efficient and robust to model misspecification on nuisance variables. Numerical studies have shown very competitive performance for our proposed methods.

A potential extension of our work involves using IV to estimate treatment effects for multi-arm and continuous treatments, with the challenge lying in the generalization of Assumption 3. Another avenue is to incorporate deep neural networks to make use of their rich expressiveness for data distribution. However, the empirical performance and theoretical properties need to be formally studied. One notable limitation is the issue of extreme weights, which can arise during the estimation process and potentially lead to instability and biased results. Addressing this limitation is crucial for improving the reliability and accuracy of our method.

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

# A    Proofs

*Proof of Proposition 1.* For any $z \in \{-1, +1\}$, we have

$$\mathbb{E}[2Y|Z = z, X]$$
$$= \mathbb{E}_U\big(\mathbb{E}[2Y|Z = z, X, U]\big)$$
$$= \mathbb{E}_U\big(\mathbb{E}[Y(1 + A)|Z = z, X, U]\big) + \mathbb{E}_U\big(\mathbb{E}[Y(1 - A)|Z = z, X, U]\big)$$
$$= \mathbb{E}_U\big(\mathbb{E}[Y(1)(1 + A)|Z = z, X, U]\big) + \mathbb{E}_U\big(\mathbb{E}[Y(-1)(1 - A)|Z = z, X, U]\big)$$
$$= \mathbb{E}_U\big(\mathbb{E}[Y(1) + Y(-1)|Z = z, X, U]\big) + \mathbb{E}_U\big(\mathbb{E}[AY(1) - AY(-1)|Z = z, X, U]\big)$$
$$= \mathbb{E}_U\big(\mathbb{E}[Y(1) + Y(-1)|X, U]\big) + \mathbb{E}_U\big(\mathbb{E}[Y(1) - Y(-1)|X, U]\mathbb{E}[A|Z = z, X, U]\big)$$

Evaluate the above equality at $z = 1$ and $z = -1$, and take the difference. Then we have

$$2\mathbb{E}[Y|Z = 1, X] - 2\mathbb{E}[Y|Z = -1, X]$$
$$= \mathbb{E}_U\Big[\mathbb{E}[Y(1) - Y(-1)|X, U]\big(\mathbb{E}[A|Z = 1, X, U] - \mathbb{E}[A|Z = -1, X, U]\big)\Big]$$

Based on Assumption 3, we have

$$\mathbb{E}[A|Z = 1, X, U] - \mathbb{E}[A|Z = -1, X, U] = \mathbb{E}[A|Z = 1, X] - \mathbb{E}[A|Z = -1, X].$$

Combining the above two, we have

$$\mathbb{E}[Y|Z = 1, X] - \mathbb{E}[Y|Z = -1, X]$$
$$= \frac{1}{2}\mathbb{E}_U\big(\mathbb{E}[Y(1) - Y(-1)|X, U]\big)\big(\mathbb{E}[A|Z = 1, X] - \mathbb{E}[A|Z = -1, X]\big)$$
$$= \mathbb{E}[Y(1) - Y(-1)|X]\big(\mathrm{P}[A|Z = 1, X] - \mathrm{P}[A|Z = -1, X]\big)$$

The above equality is equivalent to Equation (1). On the other hand, for any $x \in \mathcal{X}$,

$$\mathbb{E}\left[\frac{ZY}{\pi_Z(Z, X)}\bigg|X = x\right]$$
$$= \pi_Z(1, x)\mathbb{E}\left[\frac{Y}{\pi_Z(1, X)}\bigg|Z = 1, X = x\right] + \pi_Z(-1, x)\mathbb{E}\left[\frac{-Y}{\pi_Z(-1, X)}\bigg|Z = -1, X = x\right]$$
$$= \mathbb{E}[Y|Z = 1, X = x] - \mathbb{E}[Y|Z = -1, X = x]$$

Dividing both sides of the above equality by $\delta(x) = \mathrm{P}[A|Z = 1, X = x] - \mathrm{P}[A|Z = -1, X = x]$ yields Equation (2). □

**Lemma 3.** *Let $\ell(X, f) = \mathbb{E}[Q(X, f)|X]$ and $L(f) = \mathbb{E}\ell(X, f)$. Denote $f^* \in \mathrm{argmin}_f \ell(X, f)$. Then $f^* \in \mathrm{argmin}_f L(f)$.*

*Proof.* Denote $f^+ \in \mathrm{argmin}_f L(f)$. Then by definition, we have the following two inequalities:

$$L(f^+) \leq L(f^*) = \mathbb{E}\ell(X, f^*)$$
$$L(f^*) = \mathbb{E}\ell(X, f^*) \leq \mathbb{E}\ell(X, f^+) = L(f^+)$$

Therefore, $L(f^*) = L(f^+)$ and $f^* \in \mathrm{argmin}_f L(f)$. □

*Proof of Lemma 1.* Let $\ell(X, f) = \mathbb{E}\left[\frac{1}{\pi_Z(Z,X)}\left(\frac{2YZ}{\delta(X)} - f(X)\right)^2\bigg|X\right]$. By Lemma 3, it suffices to show $\Delta \in \mathrm{argmin}_f \ell(X, f)$.

The gradient of $\ell(X, f)$ with respect to $f$ is given by

$$\frac{\partial}{\partial f}\ell(X, f) = \mathbb{E}\left[\frac{\partial}{\partial f}\frac{1}{\pi_Z(Z, X)}\left(\frac{2YZ}{\delta(X)} - f(X)\right)^2\bigg|X\right]$$
$$= -2\mathbb{E}\left[\frac{1}{\pi_Z(Z, X)}\left(\frac{2YZ}{\delta(x)} - f(X)\right)\bigg|X\right]$$
$$= 2\mathbb{E}\left[\frac{f(X)}{\pi_Z(Z, X)}\bigg|X\right] - 2\mathbb{E}\left[\frac{2YZ}{\delta(X)\pi_Z(Z, X)}\bigg|X\right]$$
$$= 4f(X) - 4\Delta(X)$$

Since $\ell(X, f)$ is convex, we have $\Delta \in \mathrm{argmin}_f \ell(X, f)$. □

*Proof of Lemma 2.* Let $\ell_g(f) = \mathbb{E}\left[\frac{1}{\pi_Z(Z,X)}\left(\frac{2(Y-g(X))Z}{\delta(X)} - f(X)\right)^2 \Big| X\right]$. By Lemma 3, it suffices to show that for any $g$, $\arg\min_f \ell_g(X,f) = \arg\min_f \ell(X,f)$. For any $g(x)$,

$$
\begin{aligned}
\ell_g(X,f) &= \mathbb{E}\left[\frac{1}{\pi_Z(Z,X)}\left(\frac{2YZ}{\delta(X)} - f(X) - \frac{2g(X)Z}{\delta(X)}\right)^2 \Big| X\right] \\
&= \ell(X,f) + 2\mathbb{E}\left[\frac{1}{\pi_Z(Z,X)}\left(\frac{2YZ}{\delta(X)} - f(X)\right)\left(\frac{2g(X)Z}{\delta(X)}\right)\Big| X\right] \\
&\quad + \mathbb{E}\left[\frac{1}{\pi_Z(Z,X)}\left(\frac{2g(X)Z}{\delta(X)}\right)^2\Big| X\right] \\
&= \ell(X,f) + 8\frac{g(X)}{(\delta(X))^2}\mathbb{E}\left[\frac{Y}{\pi_Z(Z,X)}\Big| X\right] - 4\frac{g(X)f(X)}{\delta(X)}\mathbb{E}\left[\frac{Z}{\pi_Z(Z,X)}\Big| X\right] \\
&\quad + 4\left[\frac{g(X)}{\delta(X)}\right]^2 \mathbb{E}\left[\frac{1}{\pi_Z(Z,X)}\Big| X\right]
\end{aligned}
$$

Here the second term and the fourth term don't depend on $f$, and the third term is 0 because $\mathbb{E}\left[\frac{Z}{\pi_Z(Z,X)}\Big| X = x\right] = 0$. Therefore, $\arg\min_f L_g(f) = \arg\min_f L(f)$. $\qquad\square$

*Proof of Theorem 1.* For any $g(x)$, the variance of the derivative of the weighted loss at $f = \Delta$ is given by

$$
\begin{aligned}
&\mathrm{Var}\left(\frac{1}{\pi_Z(Z,X)}\left(\frac{2(Y-g(X))Z}{\delta(X)} - \Delta(X)\right)\right) \\
&= \mathbb{E}\left(\mathbb{E}\left[\frac{1}{(\pi_Z(Z,X))^2}\left(\frac{2(Y-g(X))}{\delta(X)} - Z\Delta(X)\right)^2\Big| X\right]\right) \\
&\triangleq \mathbb{E}[S(X,g)]
\end{aligned}
$$

Set the gradient of $S$ with respect to $g$ equal to 0. Then for any $x \in \mathcal{X}$, we have

$$
\begin{aligned}
0 &= \mathbb{E}\left[-\frac{4}{\delta(x)(\pi_Z(Z,x))^2}\left(\frac{2(Y-g(x))}{\delta(x)} - Z\Delta(x)\right)\Big| X = x\right] \\
2g(x)\mathbb{E}\left[\frac{1}{(\pi_Z(Z,x))^2}\Big| X = x\right] &= 2\mathbb{E}\left[\frac{Y}{(\pi_Z(Z,x))^2}\Big| X = x\right] - \mathbb{E}\left[\frac{Z\delta(x)\Delta(x)}{(\pi_Z(Z,x))^2}\Big| X = x\right] \\
2g(x)\left(\pi_Z^{-1}(1,x) + \pi_Z^{-1}(-1,x)\right) &= \frac{2\mu_1^Y(x)}{\pi_Z(1,x)} + \frac{2\mu_{-1}^Y(x)}{\pi_Z(-1,x)} \\
&\quad - \left(\mu_1^Y(x) - \mu_{-1}^Y(x)\right)\left(\pi_Z^{-1}(1,x) - \pi_Z^{-1}(-1,x)\right) \\
2g(x)\left(\pi_Z^{-1}(1,x) + \pi_Z^{-1}(-1,x)\right) &= \left(\mu_1^Y(x) + \mu_{-1}^Y(x)\right)\left(\pi_Z^{-1}(1,x) + \pi_Z^{-1}(-1,x)\right) \\
g(x) &= \frac{1}{2}\left(\mu_1^Y(x) + \mu_{-1}^Y(x)\right) \\
&= \frac{1}{2}\mathbb{E}\left[\frac{Y}{\pi_Z(Z,X)}\Big| X = x\right]
\end{aligned}
$$

Additionally, $S$ is convex since $\frac{\partial^2 S}{\partial g^2} = \frac{8}{(\delta(x))^2 \pi_Z(1,x)\pi_Z(-1,x)}$. By Lemma 3, $g^* \in \arg\min_g \mathbb{E}[S(X,g)]$. $\qquad\square$

*Proof of Theorem 2.* Let $\tilde{\ell}_g(X,f) = \mathbb{E}\left[\frac{1}{\pi_Z(Z,X)}\left(\frac{2(Y-\tilde{g}(X))Z}{\hat{\delta}(X)} - f(X)\right)^2 \Big| X\right]$. By Lemma 3, it suffices to show $\Delta \in \arg\min_f \tilde{\ell}_g(X,f)$, if either $\tilde{\pi}_Z = \pi_Z$ almost surely or $\tilde{g} = g$ almost surely.

The gradient of $\tilde{\ell}_g(x, f)$ with respect to $f$ is given by

$$\frac{\partial \tilde{\ell}_g(x, f)}{\partial f} = 2\mathbb{E}\left[\frac{1}{\tilde{\pi}_Z(Z, X)}f(X)\Big|X = x\right] - 2\mathbb{E}\left[\frac{1}{\tilde{\pi}_Z(Z, X)}\frac{2(Y - \tilde{g}(X))Z}{\hat{\delta}(X)}\Big|X = x\right]$$

$$= 2f(x)\left(\frac{\pi_Z(1, x)}{\tilde{\pi}_Z(1, x)} + \frac{\pi_Z(-1, x)}{\tilde{\pi}_Z(-1, x)}\right)$$

$$- \frac{4}{\hat{\delta}(x)}\left[\frac{\pi_Z(1, x)}{\tilde{\pi}_Z(1, x)}\left(\mu_1^Y(x) - \tilde{g}(x)\right) - \frac{\pi_Z(-1, x)}{\tilde{\pi}_Z(-1, x)}\left(\mu_{-1}^Y(x) - \tilde{g}(x)\right)\right]$$

If $\tilde{\pi}_Z = \pi_Z$ almost surely, then

$$\frac{\partial \tilde{\ell}_g(x, f)}{\partial f} = 4f(x) - \frac{4}{\hat{\delta}(x)}\left(\mu_1^Y(x) - \mu_{-1}^Y(x)\right) = 4\Big(f(x) - \Delta(X)\Big)$$

If $\tilde{g} = g$ almost surely, then $\tilde{g}(x) = [\mu_1^Y(x) + \mu_{-1}^Y(x)]/2$. Thus, we have

$$\frac{\partial \tilde{\ell}_g(x, f)}{\partial f} = 2f(x)\left(\frac{\pi_Z(1, x)}{\tilde{\pi}_Z(1, x)} + \frac{\pi_Z(-1, x)}{\tilde{\pi}_Z(-1, x)}\right)$$

$$- \frac{2}{\hat{\delta}(x)}\left(\frac{\pi_Z(1, X)}{\tilde{\pi}_Z(1, x)}\left(\mu_1^Y(x) - \mu_{-1}^Y(x)\right)\right)$$

$$- \frac{2}{\hat{\delta}(x)}\left(\frac{\pi_Z(-1, x)}{\tilde{\pi}_Z(-1, x)}\left(\mu_1^Y(x) - \mu_{-1}^Y(x)\right)\right)$$

$$= 2\left(f(x) - \Delta(x)\right)\left(\frac{\pi_Z(1, x)}{\tilde{\pi}_Z(1, x)} + \frac{\pi_Z(-1, x)}{\tilde{\pi}_Z(-1, x)}\right)$$

Check that $\frac{\pi_Z(1,x)}{\tilde{\pi}_Z(1,x)} + \frac{\pi_Z(-1,x)}{\tilde{\pi}_Z(-1,x)} > 0$. By convexity of $\tilde{\ell}_g(x, f)$, $\Delta \in \operatorname{argmin}_f \tilde{\ell}_g(X, f)$ in both cases. $\qquad\square$

*Proof of Theorem 3.* Let $\tilde{\ell}_h(X, f) = \left[\frac{1}{\tilde{\pi}_Z(Z,X)}\left(\frac{2(Y - \tilde{h}(X,A,Z))Z}{\tilde{\delta}(X)} - f(X)\right)^2\Big|X\right]$. By convexity of $\tilde{\ell}_h$ and Lemma 3, it suffices to show that the gradient of $\tilde{\ell}_h(x, f)$ with respect to $f$ is 0 at $f = \Delta$ in all cases. The gradient is given by

$$\frac{\partial \tilde{\ell}_h(x, f)}{\partial f} = 2\mathbb{E}\left[\frac{1}{\tilde{\pi}_Z(Z, X)}f(X)\Big|X = x\right] - 2\mathbb{E}\left[\frac{2Z}{\tilde{\pi}_Z(Z, X)}\frac{Y - \tilde{h}(X, A, Z)}{\tilde{\delta}(X)}\Big|X = x\right]$$

$$= 2f(x)\left(\frac{\pi_Z(1, x)}{\tilde{\pi}_Z(1, x)} + \frac{\pi_Z(-1, x)}{\tilde{\pi}_Z(-1, x)}\right)$$

$$- \frac{4}{\tilde{\delta}(x)}\frac{\pi_Z(1, x)}{\tilde{\pi}_Z(1, x)}\left(\mu_1^Y(x) - \mathbb{E}[\tilde{h}(X, A, Z)|Z = 1, X = x]\right)$$

$$+ \frac{4}{\tilde{\delta}(x)}\frac{\pi_Z(-1, x)}{\tilde{\pi}_Z(-1, x)}\left(\mu_{-1}^Y(x) - \mathbb{E}[\tilde{h}(X, A, Z)|Z = -1, X = x]\right)$$

- If $\tilde{\pi}_Z = \pi_Z$ almost surely, then we have the unweighted difference $\mathbb{E}[\tilde{h}(X, A, Z)|Z = 1, X = x] - \mathbb{E}[\tilde{h}(X, A, Z)|Z = -1, X = x] = 0$ by Equation (4). The resulting gradient is

$$\frac{\partial \tilde{\ell}_h(x, f)}{\partial f} = 4f(x) - \frac{2(\mu_1^A(x) - \mu_{-1}^A(x))}{\tilde{\delta}(x)}\left(\Delta(x) - \tilde{\Delta}(x)\right) - 4\tilde{\Delta}(x)$$

$$= 4\left[f(x) - \tilde{\Delta}(x) - \frac{\delta(x)}{\tilde{\delta}(x)}\left(\Delta(x) - \tilde{\Delta}(x)\right)\right]$$

It will yield $4(f(x) - \Delta(x))$ if either $\tilde{\Delta} = \Delta$ or $\tilde{\delta} = \delta$ almost surely.

- If $\tilde{\mu}_1^Y = \mu_1^Y$, $\tilde{\mu}_1^A = \mu_1^A$ and $\tilde{\Delta} = \Delta$ almost surely, and the choice of residualization function is $\tilde{h}_1$, then we have

$$\frac{\partial \tilde{\ell}_h(x,f)}{\partial f}$$

$$= 2f(x)\left(\frac{\pi_Z(1,x)}{\tilde{\pi}_Z(1,x)} + \frac{\pi_Z(-1,x)}{\tilde{\pi}_Z(-1,x)}\right) - \frac{\pi_Z(1,x)}{\tilde{\pi}_Z(1,x)}2\tilde{\Delta}(x)$$

$$+ \frac{4}{\tilde{\delta}(x)}\frac{\pi_Z(-1,x)}{\tilde{\pi}_Z(-1,x)}\left(\Delta(x)\delta(x) - [2\delta(x) - \tilde{\delta}(x)]\tilde{\Delta}(x)/2\right)$$

$$= 2(f(x) - \tilde{\Delta}(x))\left(\frac{\pi_Z(1,x)}{\tilde{\pi}_Z(1,x)} + \frac{\pi_Z(-1,x)}{\tilde{\pi}_Z(-1,x)}\right) + \frac{\pi_Z(-1,x)}{\tilde{\pi}_Z(-1,x)}\frac{\delta(x)}{\tilde{\delta}(x)}4(\Delta(x) - \tilde{\Delta}(x))$$

$$= 2(f(x) - \tilde{\Delta}(x))\left(\frac{\pi_Z(1,x)}{\tilde{\pi}_Z(1,x)} + \frac{\pi_Z(-1,x)}{\tilde{\pi}_Z(-1,x)}\right)$$

- If $\tilde{\mu}_{-1}^Y = \mu_{-1}^Y$, $\tilde{\mu}_{-1}^A = \mu_{-1}^A$ and $\tilde{\Delta} = \Delta$ almost surely, and the choice of residualization function is $\tilde{h}_2$, then we have

$$\frac{\partial \tilde{\ell}_h(x,f)}{\partial f}$$

$$= 2f(x)\left(\frac{\pi_Z(1,x)}{\tilde{\pi}_Z(1,x)} + \frac{\pi_Z(-1,x)}{\tilde{\pi}_Z(-1,x)}\right)$$

$$- \frac{4}{\tilde{\delta}(x)}\frac{\pi_Z(1,x)}{\tilde{\pi}_Z(1,x)}\left(\Delta(x)\delta(x) - [2\delta(x) - \tilde{\delta}(x)]\tilde{\Delta}(x)/2\right)$$

$$+ \frac{\pi_Z(-1,x)}{\tilde{\pi}_Z(-1,x)}2\tilde{\Delta}(x)$$

$$= 2(f(x) - \tilde{\Delta}(x))\left(\frac{\pi_Z(1,x)}{\tilde{\pi}_Z(1,x)} + \frac{\pi_Z(-1,x)}{\tilde{\pi}_Z(-1,x)}\right) - \frac{\pi_Z(1,x)}{\tilde{\pi}_Z(1,x)}\frac{\delta(x)}{\tilde{\delta}(x)}4(\Delta(x) - \tilde{\Delta}(x))$$

$$= 2(f(x) - \tilde{\Delta}(x))\left(\frac{\pi_Z(1,x)}{\tilde{\pi}_Z(1,x)} + \frac{\pi_Z(-1,x)}{\tilde{\pi}_Z(-1,x)}\right)$$

- If $(\tilde{\mu}_1^Y + \tilde{\mu}_{-1}^Y)/2 = (\mu_1^Y + \mu_{-1}^Y)/2$ and $(\tilde{\mu}_1^A + \tilde{\mu}_{-1}^A)/2 = (\mu_1^A + \mu_{-1}^A)/2$ almost surely, and the choice of residualization function is $\tilde{h}_3$, then the gradient is

$$\frac{\partial \tilde{\ell}_h(x,f)}{\partial f} = 2f(x)\left(\frac{\pi_Z(1,x)}{\tilde{\pi}_Z(1,x)} + \frac{\pi_Z(-1,x)}{\tilde{\pi}_Z(-1,x)}\right)$$

$$- \frac{4}{\tilde{\delta}(x)}\frac{\pi_Z(1,x)}{\tilde{\pi}_Z(1,x)}\left(\frac{\Delta(x)\delta(x)}{2} - (\delta(x) - \tilde{\delta}(x))\tilde{\Delta}(x)/2\right)$$

$$+ \frac{4}{\tilde{\delta}(x)}\frac{\pi_Z(-1,x)}{\tilde{\pi}_Z(-1,x)}\left(-\frac{\Delta(x)\delta(x)}{2} - (-\delta(x) + \tilde{\delta}(x))\tilde{\Delta}(x)/2\right)$$

$$= 2\left(f(x) - \tilde{\Delta}(x) - \frac{\delta(x)}{\tilde{\delta}(x)}(\Delta(x) - \tilde{\Delta}(x))\right)\left(\frac{\pi_Z(1,x)}{\tilde{\pi}_Z(1,x)} + \frac{\pi_Z(-1,x)}{\tilde{\pi}_Z(-1,x)}\right)$$

It will yield $2\left(f(x) - \Delta(x))\right)\left(\frac{\pi_Z(1,x)}{\tilde{\pi}_Z(1,x)} + \frac{\pi_Z(-1,x)}{\tilde{\pi}_Z(-1,x)}\right)$ if either $\tilde{\Delta} = \Delta$ or $\tilde{\delta} = \delta$ almost surely.

$\square$

## B   Optimal residualization (linear model example)

Consider linear model for $\Delta(x)$ with coefficients denoted by $\beta$. The objective function with outcome residualized by a function $g(x)$ is defined as follows:

$$L_g(y, z, x, \beta) = \frac{1}{\pi_Z(z, x)} \left( \frac{2(y - g(x))z}{\delta(x)} - x^T\beta \right)^2$$

Let $\beta^*$ be the unique minimizer of $Q(\beta) \triangleq \mathbb{E}[L_g(Y, Z, X, \beta)]$. Let $\ell_g(y, z, x, \beta)$ be the derivative of $L_g(y, z, x, \beta)$ with respect to $\beta$. Denote by $\hat{\beta}$ the root of the estimating equation $n^{-1} \sum_{i=1}^n \ell_g(Y_i, X_i, \beta) = 0$. By Bahadur representation, we have

$$\hat{\beta} - \beta^* = n^{-1} H^{-1} \sum_{i=1}^n \ell_g(Y_i, X_i, \beta^*) + o_P(n^{-1})$$

where $H$ is the second derivative of $Q(\beta)$ with respect to $\beta$ at $\beta = \beta^*$. Therefore, selecting the optimal $g$ is equivalent to minimizing the variance of $\ell_g(Y_i, X_i, \beta^*)$.

## C   Technical details for IV-RDL2

Unlike IV-RDL1, we need additional constraints on $h(x, a, z)$ to make sure the estimation for CATE remains consistent after the residualization.

**Lemma 4.** *For any measurable $h : (\mathcal{X}, \mathcal{A}, \mathcal{Z}) \to \mathbb{R}$ satisfying*

$$\mathbb{E}\left[ \frac{Zh(X, A, Z)}{\pi_Z(Z, X)} \Big| X = x \right] = 0 \tag{4}$$

*or equivalently, $\mathbb{E}[h(X, A, Z)|Z = 1, X = x] = \mathbb{E}[h(X, A, Z)|Z = -1, X = x]$, we have*

$$\Delta \in \underset{f}{\arg\min}\, \mathbb{E}\left[ \frac{1}{\pi_Z(Z, X)} \left( \frac{2(Y - h(X, A, Z))Z}{\delta(X)} - f(X) \right)^2 \right].$$

*Proof of Lemma 4.* Let $\ell_h(X, f) = \mathbb{E}\left[ \frac{1}{\pi_Z(Z,X)} \left( \frac{2(Y - h(X,A,Z))Z}{\delta(X)} - f(X) \right)^2 \Big| X \right]$. By Lemma 3, it suffices to show $\arg\min_f \ell_h(X, f) = \arg\min_f \ell(X, f)$. For any $h(x, a, z)$ satisfying Equation (4), we have

$$\begin{aligned}
\ell_h(X, f) &= \mathbb{E}\left[ \frac{1}{\pi_Z(Z, X)} \left( \frac{2YZ}{\delta(X)} - f(X) - \frac{2h(X, A, Z)Z}{\delta(X)} \right)^2 \Big| X \right] \\
&= \ell(X, f) + 2\mathbb{E}\left[ \frac{1}{\pi_Z(Z, X)} \left( \frac{2YZ}{\delta(X)} - f(X) \right) \left( \frac{2h(X, A, Z)Z}{\delta(X)} \right) \Big| X \right] \\
&\quad + \mathbb{E}\left[ \frac{1}{\pi_Z(Z, X)} \left( \frac{2h(X, A, Z)Z}{\delta(X)} \right)^2 \Big| X \right] \\
&= \ell(X, f) + \frac{8}{(\delta(X))^2} \mathbb{E}\left[ \frac{Yh(X, A, Z)}{\pi_Z(Z, X)} \Big| X \right] - \frac{4f(X)}{\delta(X)} \mathbb{E}\left[ \frac{Zh(X, A, Z)}{\pi_Z(Z, X)} \Big| X \right] \\
&\quad + \frac{4}{(\delta(X))^2} \mathbb{E}\left[ \frac{(h(X, A, Z))^2}{\pi_Z(Z, X)} \Big| X \right]
\end{aligned}$$

Here the second term and the fourth term don't depend on $f$, and the third term is 0. Therefore, $\arg\min_f \ell_h(X, f) = \arg\min_f \ell(X, f)$    $\square$

As shown in Lemma 4, the minimizer is invariant of a shift on outcome by a function $h$ that satisfies Eq. (4). Similar to the way of finding $\hat{g}^*$, we would like to find the function $h$ with the smallest variance of $[\pi_Z(Z, X)]^{-1}[2(Y - h(X, A, Z))Z\delta^{-1}(X) - f(X)]$ among all $h$ that satisfies Eq. (4).

**Theorem 4.** *Among all measurable* $h : (\mathcal{X}, A, Z) \to \mathbb{R}$ *satisfying Eq. (4), the following function minimizes* $\mathrm{Var}\left[ \frac{1}{\pi_Z(Z,X)} \left( \frac{2(Y - h(X,A,Z))Z}{\delta(X)} - f(X) \right) \right]$:

$$h^*(x, a, z) = \mu^Y(x) + \frac{\Delta(x)}{2}\left(a - \mu^A(x) - z\delta(x)\right)$$

*if the conditional means* $\mu^Y(x)$ *and* $\mu^A(x)$ *is one of these three pairs: (1)* $\mu_1^Y(x)$ *and* $\mu_1^A(x)$; *(2)* $\mu_{-1}^Y(x)$ *and* $\mu_{-1}^A(x)$; *(3)* $m^Y(x) \triangleq (\mu_1^Y(x) + \mu_{-1}^Y(x))/2$ *and* $m^A(x) \triangleq (\mu_1^A(x) + \mu_{-1}^A(x))/2$.

*Proof of Theorem 4.* For any $h(x, a, z)$ satisfying Equation (4), the variance of the derivative of the weighted loss $L_h(f)$ at $f = \Delta$ is given by

$$\mathrm{Var}\left( \frac{1}{\pi_Z(Z,X)} \left( \frac{2(Y - h(X,A,Z))Z}{\delta(X)} - \Delta(X) \right) \right)$$

$$= \mathbb{E}\left( \mathbb{E}\left[ \frac{1}{(\pi_Z(Z,X))^2} \left( \frac{2(Y - h(X,A,Z))}{\delta(X)} - Z\Delta(X) \right)^2 \Big| Z, X \right] \right)$$

$$\triangleq \mathbb{E}[S(X, Z, h)]$$

where

$$S(x, z, h) = \mathbb{E}\left[ \frac{1}{(\pi_Z(Z,X))^2} \left( \frac{2(Y - h(X,A,Z))}{\delta(X)} - Z\Delta(X) \right)^2 \Big| Z = z, X = x \right]$$

Now we seek to minimize $\mathbb{E}[S(X, Z, h)]$. By convexity of $S(X, Z, h)$ and Lemma 3, it suffices to show that the gradient of $S(X, Z, h)$ with respect to $h$ is 0 at $f = \Delta$, if $h$ is one of the three equivalent forms. To this end, set the gradient of $S$ with respect to $h$ to be 0. Then for any $(x, z) \in (\mathcal{X}, \mathcal{Z})$, we have

$$\mathbb{E}\left[ -\frac{4}{\delta(X)} \frac{1}{(\pi_Z(Z,X))^2} \left( \frac{2(Y - h(X,A,Z))}{\delta(X)} - Z\Delta(X) \right) \Big| Z = z, X = x \right] = 0,$$

which leads to the following condition on the optimal $h$:

$$\mathbb{E}[h(X, A, Z)|Z = z, X = x] = \mathbb{E}[Y|Z = z, X = x] - z\Delta(x)\delta(x)/2 \tag{5}$$

Since $\delta(x)\Delta(x) = \mu_1^Y(x) - \mu_{-1}^Y(x)$, it can be verified that Equation (5) implies $\mathbb{E}[h(X, A, Z)|Z = 1, X = x] = \mathbb{E}[h(X, A, Z)|Z = -1, X = x]$, which is equivalent to Equation (4). We will then verify that the following three equivalent functions satisfy Equation (5).

$$h_1^*(x, a, z) = \mu_1^Y(x) + \frac{\Delta(x)}{2}\left(a - \mu_1^A(x) - z\delta(x)\right)$$

$$h_2^*(x, a, z) = \mu_{-1}^Y(x) + \frac{\Delta(x)}{2}\left(a - \mu_{-1}^A(x) - z\delta(x)\right)$$

$$h_3^*(x, a, z) = \frac{\mu_1^Y(x) + \mu_{-1}^Y(x)}{2} + \frac{\Delta(x)}{2}\left(a - \frac{\mu_1^A(x) + \mu_{-1}^A(x)}{2} - z\delta(x)\right)$$

To see their equivalence, notice that $\Delta(x) = 2[\mu_1^Y(x) - \mu_{-1}^Y(x)]/[\mu_1^A(x) - \mu_{-1}^A(x)]$. Then we have

$$\mu_{-1}^Y(x) - \mu_{-1}^A(x)\Delta(x)/2 = \frac{\mu_{-1}^Y(x)\mu_1^A(x) - \mu_1^Y(x)\mu_{-1}^A(x)}{\mu_1^A(x) - \mu_{-1}^A(x)} = \mu_1^Y(x) - \mu_1^A(x)\Delta(x)/2,$$

and $h_3^*$ is simply the average of $h_1^*$ and $h_2^*$. It suffices to show $h_1^*$ satisfies (5). We have

$$\mathbb{E}[h_1^*(X, A, Z)|Z = 1, X = x] = \mu_1^Y(x) - \frac{\Delta(x)}{2}\delta(x)$$

$$\mathbb{E}[h_1^*(X, A, Z)|Z = -1, X = x] = \mu_1^Y(x) + \frac{\Delta(x)}{2}\left(\mu_{-1}^A(x) - \mu_1^A(x) + \delta(x)\right)$$

$$= \mu_{-1}^Y(x) + \frac{\Delta(x)}{2}\delta(x),$$

which completes the proof. □

# D Additional Simulations

In this section, we conducted simulations that evaluate the performance of the proposed framework against model mispecification on the nuisance variables. The data is generated by the same model as Setting 1 in Section 6, except that $\pi_Z(1, X) = \text{expit}\{2X_1\}$. Based on the true model, the conditional mean outcome is non-linear on $X$. However, in Setting 3, we will use its OLS estimate as a case of misspecification. In Setting 4, we deliberately used a wrong propensity $\hat{\pi}_Z(1, x) = 1/2$. We keep all the other procedures the same as Setting 1. The results are summarized in Table D. We can observe that the residualized version have superior performance, and have significant lower MSE compared to the original version.

Table 2: Simulation results: mean$\times 10^{-2}$(SE$\times 10^{-2}$). IPW-MR: the multiply robust weighted learning; BART: Bayesian additive regression trees; RD: robust direct learning; CF: causal forest. The empirical maximum value is 0.967 for setting 3 and 0.979 for setting 4.

|   |       | BART | RD | IPW-MR | CF | MRIV | IV-DL | IV-RDL1 | IV-RDL2 |
|---|-------|------|-----|--------|-----|------|-------|---------|---------|
| 3 | MSE   | $136_{(4.3)}$ | $93_{(5.7)}$ | NA | $96.6_{(1.9)}$ | $552_{(86)}$ | $194_{(14)}$ | $\mathbf{80}_{(7.0)}$ | $\underline{81.8}_{(6.5)}$ |
|   | AR    | $63.4_{(0.9)}$ | $76.2_{(0.8)}$ | $79.5_{(0.7)}$ | $76.9_{(0.8)}$ | $77.1_{(0.7)}$ | $80.1_{(0.6)}$ | $\mathbf{84.5}_{(0.6)}$ | $\underline{82.8}_{(0.7)}$ |
|   | Value | $73.7_{(1.2)}$ | $85.5_{(0.9)}$ | $89.4_{(0.8)}$ | $85.6_{(0.8)}$ | $85.6_{(0.7)}$ | $89.3_{(0.5)}$ | $\mathbf{93}_{(0.4)}$ | $\underline{91.6}_{(0.6)}$ |
| 4 | MSE   | $137_{(4.3)}$ | $86.2_{(2.6)}$ | NA | $96.7_{(1.9)}$ | $79.5_{(2.4)}$ | $103_{(5)}$ | $\mathbf{44.1}_{(3.1)}$ | $\underline{60.7}_{(3)}$ |
|   | AR    | $63.5_{(0.9)}$ | $72.1_{(0.6)}$ | $\underline{78.6}_{(1.0)}$ | $77_{(0.8)}$ | $75.6_{(0.6)}$ | $74.1_{(0.8)}$ | $\mathbf{83.6}_{(0.8)}$ | $\underline{78.5}_{(0.9)}$ |
|   | Value | $74.5_{(1.2)}$ | $84.9_{(0.6)}$ | $87.3_{(0.8)}$ | $86.3_{(0.8)}$ | $84.8_{(0.7)}$ | $83.5_{(0.9)}$ | $\mathbf{92.6}_{(0.7)}$ | $\underline{87.9}_{(0.9)}$ |

# E 3D plots for the data analysis

In the data analysis, we construct a 3-dimensional plot for the estimated CATE based on the three splitting variables (age of mom at census, age of mom at first birth, and income of father). The plot is presented in two rotations in Figure 4. The points in the plots are color-coded by the estimated CATE with red indicating more likely to work and blue indicating more likely to not work. We can see that, overall, blue points are at the bottom of the plots, with a majority of them below \$25k/year. Subgroup 3 of young mothers with extremely low fathers' income only accounts for 3% of the data and hence is hard to see here.

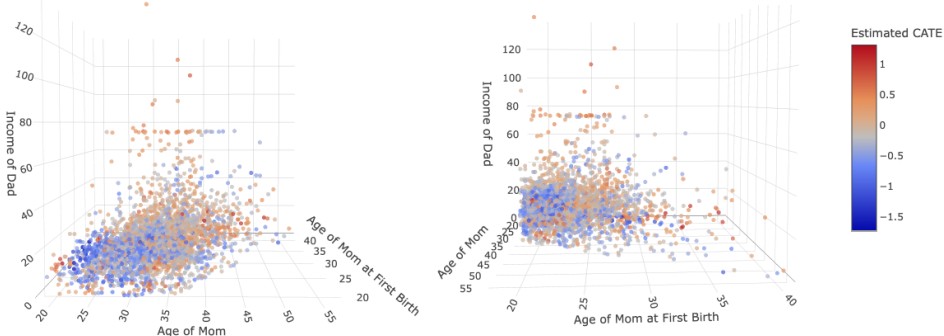

Figure 4: 3D scatter plots of three covariates colored by estimated CATE for 3000 randomly selected subjects. Both plots reflect different rotations.

