# OpenReview forum: "A Non-parametric Direct Learning Approach to Heterogeneous Treatment Effect Estimation under Unmeasured Confounding"
_NeurIPS.cc/2024/Conference — NeurIPS 2024 poster_

### Official Review · Reviewer_iUwm · 2024-07-06

**Soundness:** 2
**Presentation:** 2
**Contribution:** 2
**Rating:** 4
**Confidence:** 5

**Summary:**

In this paper, the authors proposed a general framework for estimating CATE with a possible unmeasured confounder using instrumental variables. They construct estimators that exhibit efficiency and robustness against various scenarios of model misspecification. The efficacy of the proposed framework is demonstrated through simulation studies and a real data example.

**Strengths:**

Strengths: The robustness and efficacy of the proposed method are shown in theory and simulations.

**Weaknesses:**

Please refer to Questions.

**Questions:**

The authors might consider scenarios where one might be interested in heterogeneous treatment effect on $X'$, where $X'$ is a subset of $X$.

**Limitations:**

Please refer to Questions.

---

> ### Author Rebuttal · Authors · 2024-08-06
>
> We appreciate your feedback and the time you have taken to review our paper. However, we are unsure about the question being raised in this comment. Our goal is to estimate the conditional average treatment effect of $A$ on $Y$ given $X$. We are unsure what you meant by heterogeneous treatment effect "on a subset of $X$". In this paper, we do not consider the treatment effect on "$X$". If you meant treatment effect of $A$ on $Y$ given a subset of $X$, we note that the incorporation of a variable selection method under our framework is very straightforward given that the final step of our method is a weighted least square regression problem. If you meant the treatment effect on a subset of $Y$, not of $X$, we note that, currently, we consider $Y$ to be univariate.

---

### Official Review · Reviewer_M1zT · 2024-07-10

**Soundness:** 2
**Presentation:** 3
**Contribution:** 3
**Rating:** 5
**Confidence:** 2

**Summary:**

The author mainly introduces a method of Directly Learning using Instrumental Variables (IV-DL) to estimate the conditional average treatment effect (CATE) $\Delta(x)$ and optimal Individualized Treatment Regime (ITR) $\hat{d(x)}$ in the presence of unobserved confounding. They propose two efficient and robust estimators, IV-RDL1 and IV-RDL2, by residualizing the outcome. The authors conduct two simulation settings and use real-world data to demonstrate the efficiency of the approach.

**Strengths:**

This paper primarily focuses on estimating conditional average treatment effects and determining the optimal individualized treatment regime using the Direct Learning with Instrumental Variable Approach. It presents a well-structured logical framework to discuss this concept.

**Weaknesses:**

I think overall the authors did interesting research, but my main concerns are listed below.

The variables A, Y, and Z in this paper are all binary variables. It would be beneficial to discuss how the framework of IV-DL can be extended or adapted to handle a continuous instrumental variable 𝑍. The current focus on binary variables may limit the generalizability of the findings.

In Section 5, the authors do not provide a detailed introduction to work similar to IV-RDL2. A more thorough explanation of the differences and connections between this work and related research would enhance the reader's understanding of the unique contributions and context of the presented study.

**Questions:**

1. In section 6, why is the performance of IV-RDL1 than IV-RDL2 in three metrics?

2. The authors should conduct some experiments to show whether IV-RDL1 and IV-RDL-2 are robust compared to other methods in the presence of misestimation.

3. The authors miswrite “$f(x)=\tilde{x}^{T}\mathbf{\beta}$” as “$\Delta(x)=\tilde{x}^{T}\mathbf{\beta}$” in line 152.

**Limitations:**

The authors discuss some of the algorithm's shortcomings.

---

> ### Author Rebuttal · Authors · 2024-08-06
>
> We thank the reviewer for the thoughtful and constructive feedback on our paper. We have carefully considered each comment and will make revisions to address the concerns raised. Below, we provide detailed responses to the reviewer's comments, along with descriptions of the changes that will be made to the paper to improve its overall quality.
>
> ## Restricted to the binary case
> We acknowledge the reviewer's observation that the current paper focuses solely on binary treatment. During our research, we discovered that extending our framework to accommodate other types of treatment is a non-trivial task. We are currently working on another project dedicated to expanding our framework to include multi-arm treatment settings. This extension is rather complicated and will be detailed in a forthcoming paper.
>
> ## Details on IV-RDL2
> Due to space constraints, we have omitted most of the details on IV-RDL2 from the main text. The remaining comparison with related work can be found in lines 257-264, specifically with MRIV by Frauen and Feuerriegel (2022). We will provide a detailed introduction and comprehensive comparison in the appendix.
>
> ## Question Section
> 1. IV-RDL2 is not necessarily superior to IV-RDL1. Both residualized versions are sensitive to the accuracy of their nuisance parameter estimates. Since IV-RDL2 requires more nuisance parameters than IV-RDL1, it is likely to introduce more variance into the CATE estimate. The utility of these residualized versions would be greatly enhanced if we possessed domain knowledge about the baseline conditional means.
> 2. Our submission did include two additional settings in Appendix D that demonstrate the robustness properties of the residualized estimators. In these examples, we intentionally employed incorrect estimates or models for the nuisance parameters. Despite this, both IV-RDL1 and IV-RDL2 exhibited superior performance compared to other methods.
> 3. We will revise this paragraph to enhance its clarity.

---

> > ### Comment · Reviewer_M1zT · 2024-08-12
> > **One quick question**
> >
> > Thank you for your response. Regarding the binary case, you mentioned, "During our research, we discovered that extending our framework to accommodate other types of treatment is a non-trivial task." Could you please provide more insight or intuition on why this is the case?

---

> > > ### Author Response · Authors · 2024-08-12
> > > **Response to Question**
> > >
> > > Thank you for your follow-up question. We appreciate your interest in understanding the challenges associated with extending our framework to accommodate other types of treatments. We will briefly highlight the key aspects of identification in the binary case and then explain the challenges involved in generalizing to other types of treatments.
> > >
> > > To begin with, we will need the following notations:
> > > - $\Delta(X)=E[Y(1)-Y(-1)\vert X]$
> > > - $\delta_Y(X)=E[Y\vert Z=1,X]-E[Y\vert Z=-1,X]$
> > > - $\delta_A(X)=P[A\vert Z=1,X]-P[A\vert Z=-1,X]$
> > > - $\tilde\delta_Y(X,U)=E[Y(1)-Y(-1)\vert X, U]$
> > > - $\tilde\delta_A(X,U)=P[A\vert Z=1,X, U]-P[A\vert Z=-1,X, U]$
> > >
> > > The proof of Proposition 1 demonstrates that, in the binary case, the identification on the Conditional Average Treatment Effect (CATE), denoted $\Delta(x)$, hinges on the following relationship:
> > > $$\delta_Y(X)=E_U[\tilde\delta_Y(X,U)\tilde\delta_A(X,U)]=\Delta(X)\delta_A(X)$$
> > > Here, Assumption 2f provides a sufficient condition for the validity of the second equation. Then we have identified the CATE: $\Delta(X)=\delta_Y(X)/\delta_A(X)$.
> > >
> > > For a $k$-arm treatment scenario, a natural approach involves selecting one treatment arm as the baseline and defining the CATE as the difference between each of the other arms and this baseline. This results in a CATE vector of dimension $k-1$. Extending Assumption 2f to accommodate this setup and maintain the equality $E_U[\delta_Y(X,U) \delta_A(X,U)] = \Delta(X) \delta(X)$ is not straightforward. In particular, this identification equation will become a system of linear equations, whose solution requires the inversion of a $(k-1)$ dimensional square matrix. Hence, we feel that this would be too complicated to incorporate into the current papers as an additional section; rather, it deserves a separate paper.
> > >
> > > The challenge increases with the generalization to continuous treatments, as the existing identification relies on differences between conditional means given two discrete IV levels. This suggests the need for novel theoretical frameworks or assumptions tailored to these more complex scenarios.

---

### Official Review · Reviewer_qaqj · 2024-07-12

**Soundness:** 3
**Presentation:** 3
**Contribution:** 3
**Rating:** 7
**Confidence:** 4

**Summary:**

The authors study the problem of estimating the conditional average treatment effect (CATE) under the assumption of unmeasured confounding. The authors focus on the specific scenario where some observed variable acts as instrument w.r.t. unmeasured confounder but might be confounded by some other observed confounder, so that standard IV methods may fail. They derive a method which extends Direct Learning (DL) by an additional scaling factor of the outcomes. This scaling factor is the CATE of the instrument on the treatment, which can be estimated with standard methods (e.g., DL). In a simulation study, the authors compare the proposed method to a set of baseline methods.

**Strengths:**

The problem and method are well-presented. The resulting method is simple but elegant. It extends Direct Learning by, first estimating the CATE of the instrument on the treatment, and estimates the CATE of the treatment on the outcome through Direct Learning leveraging the result of the first step.

The authors prove identifiability under a provided set of assumptions. They propose two ways to residualize the outcomes in order to reduce the variance of the estimator, and provide a set of sufficient conditions (in terms of correctly specified nuisance functions) under which the estimator yields consist CATE estimates.

**Weaknesses:**

Assumption 2.f seems rather strong as the unobserved confounder can only additively affect the treatment. The data generating process in the experimental section violates this assumption. As the proposed method still outperforms the baselines, this may suggest that the method is less sensitive to the assumption. Though, that should be studied empirically in more detail.

Compared to other recent publications focusing on estimating the (conditional) treatment effect, the assumed data generating process in the simulation seems overly simple. Other methods involving GPs, normalizing flows, and other highly non-linear models allow for high-dimensional confounders. They are typically assessed using semi-artificial data (e.g., with images as confounders and image labels as confounding mechanism). Without such hard problems and the corresponding baseline methods, it is hard to assess the overall practical value of the proposed method.

**Questions:**

The paper is clear; no questions but some minor comment:
- The term instrumental variable for Z might be a bit confusing as Z and Y are confounded; Z is an IV w.r.t. unmeasured confounder. It may help to clarify that in the very beginning.
- Figure 1 might be improved in terms of order and size of the nodes; having the instrument on the right, the treatment at the top, and the outcome on the left is rather unconventional.
- lines 151-164 have likely little value as this should be known to the Neurips audience

**Limitations:**

NeurIPS Paper Checklist is provided; no concerns.

---

> ### Author Rebuttal · Authors · 2024-08-06
>
> We thank the reviewer for the thoughtful and constructive feedback on our paper. We have carefully considered each comment and will make revisions to address the concerns raised. Below, we provide detailed responses to the reviewer's comments, along with descriptions of the changes that will be made to the paper to improve its overall quality.
>
> ## Assumption 2f
> Thank you for pointing this out. There is a weaker version of the assumption regarding the additive effect of unobserved confounders: either a) the effect of the unmeasured confounder on the treatment is additive, or b) the effect of the unmeasured confounder on the outcome is additive. The identification result holds if either condition a) or b) is satisfied (with assumption 2f being equivalent to b). Essentially, for identification, we need the following equation to hold:
> \begin{equation}
>     E_U\Big[E[Y(1)-Y(-1)\vert X, U]\big(E[A\vert Z=1,X, U]-E[A\vert Z=-1,X, U]\big)\Big]
>     =E[Y(1)-Y(-1)\vert X]\big(E[A\vert Z=1,X]-E[A\vert Z=-1,X]\big)
> \end{equation}
> A sufficient condition for this would be to assume that at least one of these differences does not depend on $U$. In the weaker assumption, case a) implies $E[A\vert Z=1,X, U]-E[A\vert Z=-1,X, U]=E[A\vert Z=1,X]-E[A\vert Z=-1,X]$, and case b) implies $E[Y(1)-Y(-1)\vert X, U]=E[Y(1)-Y(-1)\vert X]$. If either case is true, the above equation holds. As for the simulation study, the effect of $U$ on outcomes is additive, which corresponds to case b) in the weaker assumption. We will modify the current assumption 2f to the weaker version, including the case of either a) or b) above.
>
> ## High-dimensional confounder
> The proposed framework is compatible with a wide range of state-of-the-art machine learning methods, as it ultimately involves a weighted regression problem with the modified outcome serving as the response variable. For high-dimensional data, the implementation of more complex learning algorithms under the general framework of our method is feasible.
>
> ## Comments in the Question section
> We appreciate your suggestion and will incorporate a summary of the relationships among $Z$, $Y$, and the unmeasured confounder at the beginning of Section 2. We will also construct a new Figure 1 and shorten the paragraphs in lines 151-164.

---

### Official Review · Reviewer_M5di · 2024-07-29

**Soundness:** 3
**Presentation:** 2
**Contribution:** 2
**Rating:** 6
**Confidence:** 4

**Summary:**

This paper introduces a new type of CATE estimator using instrumental variables. The proposed method employs the direct learning approach.

**Strengths:**

1. The paper is self-contained and comprehensible.
2. Besides developing the CATE estimator, the paper also proposes an estimator for finding the optimal treatment regimes.

**Weaknesses:**

__Missing literature review__

The paper most closely related to this work is [Machine Learning Estimation of Heterogeneous Treatment Effects with Instruments[(https://proceedings.neurips.cc/paper/2019/file/3b2acfe2e38102074656ed938abf4ac3-Paper.pdf). It develops a fast-converging CATE estimator for local average treatment effects using instrumental variables. However, this paper is not cited in the literature review. Please consider including a discussion of this paper for richer context. More importantly, please compare your work with this paper to highlight the novelty of the current work.

__Weak motivation on directed learning__

The introduction section lacks plausible reasons for proposing directed learning. What are the alternative methods and their pros and cons? Why should we specifically consider the directed learning approach?

__Validity of Assumption 2__

Assumption 2f is a weaker version of the following assumption: "$U$ is noninformative to $A$ given $Z$ and $X$ (i.e., $A \perp U \mid X,Z$)." Given that there are no practical settings where Assumption 2f holds while $A \perp U \mid X,Z$ doesn't (except some peculiar parametrization), and both assumptions are non-testable, I don't see any practical distinction between $A \perp U \mid X,Z$ and Assumption 2f. In other words, Assumption 2f is just another representation of $A \perp U \mid X,Z$ tailored for identification.

Combining $A \perp U \mid X,Z$ with Assumption 2c ($Z \perp U \mid X$) results in $(U \perp A \cup Z \mid X)$ by the contraction property of conditional independence. This means $U$ does not influence $(A,Z)$ given $X$. Consequently, in any related causal graph, there should be no edges from $U$ to $A$. This leads to the ignorability condition that $Y(a) \perp A \mid X$.

In summary, interpreting Assumption 2f as $A \perp U \mid X,Z$ means Assumption 2 is essentially an ignitability assumption. Therefore, it is important to discuss the validity of Assumption 2 in practical settings, to disprove that Assumption 2f is merely another representation of $A \perp U \mid X,Z$ designed for identification. Have you considered the LATE setting, given that the estimand in Proposition 1 will remain unchanged?

__More analysis is required__

Multiple robustness properties provided in Theorem 3 imply that the proposed estimator converges to the optimal estimator faster. For example, if nuisances converge at an $n^{-1/4}$ rate, where $n$ is the number of samples, then the estimator converges at an $n^{-1/2}$ rate. These results are beneficial since they guarantee fast convergence. While Theorem 3 is attractive, it is somewhat impractical because, in practice, the working model is rarely considered a true model. Please provide more analysis on the rate of convergence concerning the convergence rate of nuisance parameters.

__Fair comparison with other estimators__

Even if the empirical evidence in Table 1 is strong, the discussion on why the proposed estimator converges faster than other multiply-robust estimators, such as MRIV, is missing. Asymptotically, there are no reasons to believe the proposed estimator converges faster than the MRIV estimator. Can you provide a discussion on why the proposed estimator converges faster than its competitors?

**Questions:**

1. $\Delta(x)$ in line 89 and $\Delta(x)$ in line 94 are the same?

2. What are the practical examples where Assumption 2 holds?

3. Is there a reason to choose Assumption 2 other than the LATE assumption? Both assumptions yield the same target parameter (in Proposition 1).

**Limitations:**

1. The paper assumes discrete/binary $Z$.

2. The paper is relying on Assumption 2.

---

> ### Author Rebuttal · Authors · 2024-08-06
>
> We thank the reviewer for the thoughtful and constructive feedback on our paper. We have carefully considered each comment and will make revisions to address the concerns raised. Below, we provide detailed responses to the reviewer's comments, along with descriptions of the changes that will be made to the paper to improve its overall quality.
>
> ## Missing literature review
> Both the method of DRIV, as proposed by Syrgkanis et al. (2019), and that of MRIV, by Frauen and Feuerriegel (2022), share some similarities with our method. Both approaches construct the pseudo-outcome using the efficient influence function (EIF) of the Average Treatment Effect (ATE). However, MRIV incorporates the multiply robust property as detailed by Wang and Tchetgen Tchetgen (2018), whereas DRIV features a doubly robust property only. Given that our method exhibits an extended multiply robust property, we have opted for a more detailed comparison with MRIV. That being said, we will include DRIV in our literature review including a more thorough comparison.
>
> ## Motivation for direct learning
> The motivation for the direct learning approach was outlined in lines 24-29, highlighting its advantages over Q-learning. A more detailed comparison is provided in lines 88-95. Under the unconfoundedness assumption, Q-learning models the conditional mean outcomes separately and calculates the estimated Conditional Average Treatment Effect (CATE) using these estimates, denoted as $\hat\Delta(X)=\hat E[Y|X, A=1]-\hat E[Y|X, A=0]$. In contrast, the Direct Learning approach models the difference between the conditional mean outcomes directly, which can be expressed as $\Delta(X)= E[Y|X, A=1]-E[Y|X, A=0]=E[AY/\pi_A(A, X)|X]$. Regarding your Question 1 about the definition of $\Delta(x)$, the two definitions in line 89 and line 94 are both transformed from the definition of CATE under the unconfoundedness assumption and are essentially the same. We will reorganize the introduction section to enhance clarity.
>
> ## Assumption 2f
> Wang and Tchetgen Tchetgen (2018) first showed that CATE is identified by the conditional Wald estimand (Eq.1 in Proposition 1) with a weaker assumption: either a) the effect of the unmeasured confounder on the treatment is additive or b) the effect of the unmeasured confounder on the outcome is additive. This result is adopted in many later works such as MRIV (Frauen and Feuerriegel, 2022), Causal Forest (Athey, Tibshirani, and Wager, 2019), and IPW-MR (Cui et al., 2021). The assumption used by Syrgkanis et al. (2019) is equivalent to the stronger version (b) and ours is equivalent to (a). An example of when Assumption 2f holds can be when the unmeasured confounder only has additive effects on $A$. In the context of the Local Average Treatment Effect (LATE), the formulation aligns with the identification result on CATE expressed in Equation 1 of Proposition 1. However, LATE focuses specifically on a subgroup of the population, whereas identifying the CATE provides a more comprehensive result. Furthermore, it can be verified that $\Delta(x)$ is equivalent to LATE under the monotonicity assumption, as discussed by Imbens and Angrist (1994).
>
> ## About "rate of convergence"
> The findings in Theorem 3 describe six scenarios where the IV-RDL2 will provide a consistent estimate of the CATE. We have not claimed a faster convergence rate compared to other methods, and currently, we lack theoretical results on convergence rates. It's important to note that both IV-RDL1 and IV-RDL2 are heavily dependent on the accurate estimation of nuisance parameters. We recommend using IV-RDL1 as it requires fewer nuisance parameters unless specific background knowledge about the distributions of the additional nuisance parameters used in IV-RDL2 is available. Studying the rate of convergence of the estimators concerning the convergence rate of the nuisance parameter is an interesting and important future work.
>
> ## Restricted to the binary case
> We acknowledge the reviewer's observation that the current paper focuses solely on binary treatment. During our research, we discovered that extending our framework to accommodate other types of treatment is a non-trivial task. We are currently working on another project dedicated to expanding our framework to include multi-arm treatment settings. This extension is rather complicated and will be detailed in a forthcoming paper.

---

> ### Comment · Reviewer_M5di · 2024-08-11
> **Response**
>
> Thank you for your response. My concerns about the motivation and the justification of Assumption 2f have been addressed. However, the following questions remain:
> 1. Can you provide a discussion on why the proposed estimator converges faster than its competitors?
> 2. What is the rate of convergence in relation to the convergence rate of nuisance parameters?
>
> I believe the paper could be stronger if
> 1. the motivation of the direct learning is more clearly explained in the introduction section, and
> 2. the rate of convergence is added.
>
> By the way, the current answer misses the response for my question: Can you provide a discussion on why the proposed estimator converges faster than its competitors?

---

> > ### Author Response · Authors · 2024-08-12
> > **Response to Comment**
> >
> > Thank you for informing us that your concerns about the motivation and justification of Assumption 2f have been resolved. We value your continued engagement and feedback. Below are our responses to the remaining questions and suggestions:
> >
> > ## Remaining Questions on Convergence Rate
> >
> > We would like to clarify that we did not claim the proposed estimator has a faster convergence rate. Our theoretical results only establish consistency. We will review our text to ensure it does not suggest that we provide convergence rate results. While we are not aware of similar findings in the existing literature, especially related to Wang and Tchetgen Tchetgen (2018), this could be an intriguing direction for future research.
> >
> > ## Suggestions to Strengthen the Paper
> >
> > 1. **Motivation for Direct Learning in Introduction:**
> >   We will move the explanation about the benefits of direct learning over traditional methods like Q-learning to the introduction section as suggested to improve the motivation.
> >
> > 2. **Including Rate of Convergence:**
> >   We agree. As mentioned, while we are not aware of similar findings in the existing literature, especially related to Wang and Tchetgen Tchetgen (2018), this could be an intriguing direction for future research.

---

> > > ### Comment · Reviewer_M5di · 2024-08-13
> > > **Response**
> > >
> > > Thank you for addressing my concern. I will raise my point from 5 to 6, given that the end of the discussion period is coming.
> > >
> > > My final question is this: Can you discuss the fast convergence shown in the experiment? In theory, MR-IV and the proposed estimators both converge fast, but the proposed estimator outperforms in the simulation. Can you justify this simulation result?

---

> > > > ### Author Response · Authors · 2024-08-13
> > > > **Response to Question**
> > > >
> > > > Thank you for your feedback and for raising your score.
> > > >
> > > > We understand that you would like some explanation for the better performance of the proposed method for the finite samples in the simulation. One possible reason is that our proposed IV-RDL1 method requires fewer nuisance estimates than MRIV. Another reason may be that in all our proposed methods, we use inverse propensity score $1/\pi_Z(Z,X)$ as the weights in a weighted least square framework. This helps to balance different IV groups and helps to reduce bias and variance (see Seaman and White, 2013).

---

### Author Rebuttal · Authors · 2024-08-06

We would like to extend our sincere gratitude for the thorough and constructive feedback on our paper. We have carefully considered all comments and will make revisions to address the concerns raised. Below, we provide a summarized response to the main comments, along with descriptions of the changes that will be made to the paper.

**Literature Review:**
We will include a more thorough comparison with DRIV, in addition to the detailed comparison with MRIV, to enhance the literature review section.

**Motivation for Direct Learning:**
The introduction will be reorganized to better highlight the advantages of direct learning over Q-learning, providing additional details to clearly distinguish the two approaches.

**Assumption 2f:**
We clarified Assumption 2f and provided examples to illustrate when it holds, aligning it with previous works by Wang and Tchetgen Tchetgen (2018). The weaker version of Assumption 2f will be included to provide additional clarity on the additive effects of unobserved confounders.


**High-Dimensional Confounder:**
Our framework is compatible with a wide range of machine learning methods, making it feasible for high-dimensional data and variable selection. This point will be emphasized to highlight the flexibility of our approach.

**Details on IV-RDL2:**
Due to space constraints, detailed information on IV-RDL2 was omitted from the main text. We will include a more detailed introduction and comprehensive comparison in the appendix to provide a clearer understanding of IV-RDL2 and its relation to related work.

**Superiority of IV-RDL2:** We clarified that IV-RDL2 is not necessarily superior to IV-RDL1, and both methods are sensitive to the accuracy of nuisance parameter estimates.

**Restricted to Binary Case:**
We acknowledge the current focus on binary treatment. We are actively working on expanding our framework to include multi-arm treatment settings. The extension is complicated and will be detailed in a forthcoming paper.

We hope these revisions address the concerns effectively and believe these changes will strengthen the paper while clarifying points of ambiguity. Thank you once again for your valuable feedback.

---

### Decision · Program_Chairs · 2024-09-25

**Decision:**

Accept (poster)

**Comment:**

This paper proposes a new CATE estimator (variants) in the instrumental variable setting using direct learning, under the setting when the instrument may be confounded by an observed variable. The paper provides identification conditions, a multiply robust estimator and conduct empirical evaluation on synthetic simulations and real world data. Theoretical results provide standard convergence guarantees of the doubly/multiply-robust estimator.

Overall all reviewers found the paper clear, comprehensible. Deriving optimal treatment regimes in addition to CATE was considered a strength.

Multiple reviewers raised some concerns about Assumption 2f, particularly in the context of existing literature of Wang and Tchetgen Tchetgen (2018) (as well as its justification of additive effect of unobserved confounders on treatment --- or outcome as was clarified). Impact of high-dimensional confounding was raised, and addressed to some extent (doubly robust estimators still have empirical limits in practice, but this is sufficiently addressed for the paper for now).

Overall I have ignored the uninformative review from iUwm and recommended accept in this case.